# Increased glutarate production by blocking the glutaryl-CoA dehydrogenation pathway and a catabolic pathway involving L-2-hydroxyglutarate

Manman Zhang[1], Chao Gao[1], Xiaoting Guo[1], Shiting Guo[1], Zhaoqi Kang[1], Dan Xiao[1], Jinxin Yan[1], Fei Tao[2], Wen Zhang[3], Wenyue Dong[4], Pan Liu[1], Chen Yang[4], Cuiqing Ma[1] & Ping Xu[2]

Glutarate is a five carbon platform chemical produced during the catabolism of L-lysine. It is known that it can be catabolized through the glutaryl-CoA dehydrogenation pathway. Here, we discover that *Pseudomonas putida* KT2440 has an additional glutarate catabolic pathway involving L-2-hydroxyglutarate (L-2-HG), an abnormal metabolite produced from 2-ketoglutarate (2-KG). In this pathway, CsiD, a $Fe^{2+}$/2-KG-dependent glutarate hydroxylase, is capable of converting glutarate into L-2-HG, and LhgO, an L-2-HG oxidase, can catalyze L-2-HG into 2-KG. We construct a recombinant strain that lacks both glutarate catabolic pathways. It can produce glutarate from L-lysine with a yield of 0.85 mol glutarate/mol L-lysine. Thus, L-2-HG anabolism and catabolism is a metabolic alternative to the glutaryl-CoA dehydrogenation pathway in *P. putida* KT2440; L-lysine can be both ketogenic and glucogenic.

[1] State Key Laboratory of Microbial Technology, Shandong University, Jinan 250100, People's Republic of China. [2] State Key Laboratory of Microbial Metabolism, Joint International Research Laboratory of Metabolic & Developmental Sciences and School of Life Sciences & Biotechnology, Shanghai Jiao Tong University, Shanghai 200240, People's Republic of China. [3] The Second Hospital of Shandong University, Jinan 250033, People's Republic of China. [4] CAS-Key Laboratory of Synthetic Biology, Shanghai Institute of Plant Physiology and Ecology, Chinese Academy of Sciences, Shanghai 200032, People's Republic of China. Correspondence and requests for materials should be addressed to C.G. (email: jieerbu@sdu.edu.cn) or to P.X. (email: pingxu@sjtu.edu.cn)

L -2-Hydroxyglutarate (L-2-HG) is a competitive inhibitor of multiple 2-ketoglutarate (2-KG)-dependent dioxygenases involved in a wide range of biological processes, including prolyl hydroxylases (PHD) and histone demethylases[1,2]. Like its mirror-image enantiomer D-2-hydroxyglutrate (D-2-HG)[3–5], L-2-HG is also viewed as an abnormal metabolite leading to pathogenesis[6,7]. Acidic pH can enhance the production of L-2-HG under the hypoxic condition in both normal and malignant cells[8]. Currently, it is generally conceded that L-2-HG is solely produced from the reduction of 2-KG by lactate dehydrogenase and malate dehydrogenase[9–14].

Glutarate is an important metabolite in animals, plants, and microbes[15–17]. It is distributed in various habitats and can be produced during the biological catabolism of several amino acids (such as L-lysine, L-hydroxylysine, and L-tryptophan) and aromatic compounds (such as nicotinate and benzoate)[17]. The only reported glutarate catabolic route is the glutaryl-coenzyme A (CoA) dehydrogenation pathway. In this classical pathway, glutarate is first converted into glutaryl-CoA[18]. Then, glutaryl-CoA dehydrogenase (GDH) catalyzes the α,β-dehydrogenation of glutaryl-CoA to glutaconyl-CoA and the decarboxylation of glutaconyl-CoA to crotonyl-CoA and carbon dioxide[19–21]. Crotonyl-CoA is subsequently converted into acetoacetate-CoA and then into two molecules of acetyl-CoA[22]. Since L-lysine can only be metabolized through acetoacetate-CoA and acetyl-CoA, it is traditionally viewed as a solely ketogenic amino acid[23].

Glutarate is also an attractive C5 platform chemical with versatile applications, especially as a monomer in the synthesis of nylon[24,25]. Today, glutarate is obtained via various chemical processes that rely on petrochemical precursors[26]. Biotechnological production of glutarate can be accomplished through the catabolism of L-lysine using recombinant Escherichia coli, an important industrial strain that lacks the GDH encoding gene. However, the yield of glutarate from L-lysine by recombinant E. coli is rather low[25,27]. These intriguing phenomena prompted us to study the other unidentified glutarate catabolic pathway(s) in nature.

In this study, we try to identify the mechanism and physiologic function of microbial L-2-HG metabolism using Pseudomonas putida KT2440, one of the rhizosphere-dwelling model organisms[28,29]. P. putida KT2440 can grow on L-lysine or L-hydroxylysine, involving glutarate as an intermediate[30–32]. Aside from the well-studied pathway requiring GDH, a glutarate catabolic pathway involved with L-2-HG anabolism and catabolism is proposed in this work. The additional glutarate catabolic pathway is composed of two key enzymes, a glutarate hydroxylase (CsiD) and an L-2-HG oxidase (LhgO) (Fig. 1a, in red). Thus, we uncover an updated version of glutarate and L-2-HG metabolism in P. putida KT2440, and reveal that L-lysine can also be viewed as a glucogenic amino acid. Additionally, the glutarate production from L-lysine with a high yield (0.85 mol glutarate/mol L-lysine) has been realized.

## Results

**GDH is not indispensable for glutarate utilization.** The glutaryl-CoA dehydrogenation pathway and its key enzyme glutaryl-CoA dehydrogenase (GDH) have been well-studied in both eukaryotes and prokaryotes[17,33,34]. In this study, we disrupted the gdh gene that encodes GDH (PP0158) in P. putida KT2440. Surprisingly, the growth of the gdh mutant (Δgdh) was not strikingly different from the wild-type strain with glutarate as the sole carbon source. The Δgdh mutant could also grow with glutarate as the sole carbon source, but with a slightly reduced growth rate and glutarate consumption rate than the wild-type (Fig. 1b), suggesting the presence of other unidentified glutarate metabolism pathway(s) in P. putida KT2440.

**CsiD and LhgO are induced during glutarate utilization.** 4-Aminobutyrate aminotransferase (GabT) and succinate-semialdehyde dehydrogenase (GabD) play important roles in the production of succinate from 4-aminobutyrate in E. coli[35]. Due to the structural similarity between 4-aminobutyrate and 5-aminovalerate, GabT and GabD might also participate in the production of glutarate from 5-aminovalerate in E. coli (Fig. 1c). It is worth noting that the genes csiD and ygaF are located upstream of gabD in E. coli[36,37] (Fig. 1c). CsiD belongs to the non-haem Fe$^{2+}$/2-KG-dependent dioxygenase family (EC 1.14.11)[38–40]. YgaF is an L-2-hydroxyglutarate (L-2-HG) oxidase that oxidizes L-2-HG to 2-ketoglutarate (2-KG)[41]. Since L-2-HG might be formed by the hydroxylation of glutarate and the csiD and ygaF are located adjacent to gabT and gabD, it is reasonable to speculate that CsiD and YgaF are involved in the utilization of glutarate.

There is a possible unidentified glutarate metabolic pathway in P. putida KT2440, and the ortholog proteins of CsiD and YgaF have also been annotated in its genome, named CsiD (PP2909) and LhgO (PP2910). Thus, we used P. putida KT2440 as a model strain to study the hypothetical role of CsiD and LhgO in glutarate utilization. The expression of csiD and lhgO during glutarate utilization was analyzed by reverse transcription-PCR (RT-PCR) experiments. Whereas no obvious RT-PCR products were observed by using mRNA from P. putida KT2440 cells grown in pyruvate, two clear RT-PCR fragments of csiD and lhgO were observed using mRNA from P. putida KT2440 cells grown in glutarate (Fig. 1d). This result suggests that the csiD and lhgO genes in P. putida KT2440 are glutarate-inducible and they might be involved in an unidentified pathway for glutarate metabolism (Fig. 1a).

**CsiD is a Fe$^{2+}$/2-KG-dependent dioxygenase acting on glutarate.** CsiD of P. putida KT2440 was cloned, expressed in E. coli BL21(DE3), and purified using a HisTrap column (Fig. 2a). Based on the results of size exclusion chromatography (Fig. 2b) and the mass of the CsiD monomer (calculated as 38.9 kDa according to its deduced amino acid sequence), the purified recombinant CsiD showed a polymerization degree of 7.85, indicating that this protein exists as an octamer.

The CsiD-dependent oxygen consumption detected by a Clark-type oxygen electrode was almost absent without the addition of 2-KG, while noticeable oxygen consumption was observed when 2-KG was added (Fig. 2c). The product of the CsiD-catalyzed oxidative decarboxylation of 2-KG was analyzed by high-performance liquid chromatography (HPLC). A compound that had a retention time of 19.37 min, which corresponded to the peak of authentic succinate, was detected (Fig. 2d). The result of liquid chromatography-tandem mass spectrometry (LC-MS/MS) further confirmed the production of succinate by CsiD (Supplementary Fig. 1a). Like other typical Fe$^{2+}$/2-KG-dependent dioxygenases, CsiD has unspecific activity toward 2-KG to produce succinate.

As shown in Fig. 2c, a much higher oxygen consumption rate was observed when glutarate and 2-KG were added in the reaction mixture, indicating that CsiD is a Fe$^{2+}$/2-KG-dependent dioxygenase capable of acting on glutarate. Substrate specificity of CsiD was then examined with 5-aminovalerate, 2-aminoadipate, 2-ketoadipate, glutarate, succinate, adipate, glutaryl-CoA, and L-lysine as the test substrates and 2-KG as the co-substrate. CsiD seems to have a narrow substrate specificity, only glutarate being rapidly oxidized by the enzyme (Fig. 2e). The activity of CsiD was almost negligible in the absence of 2-KG and strongly reduced (to 7.0%) without glutarate (Fig. 2f). Ascorbate is an established activator of Fe$^{2+}$/2-KG-dependent dioxygenases through

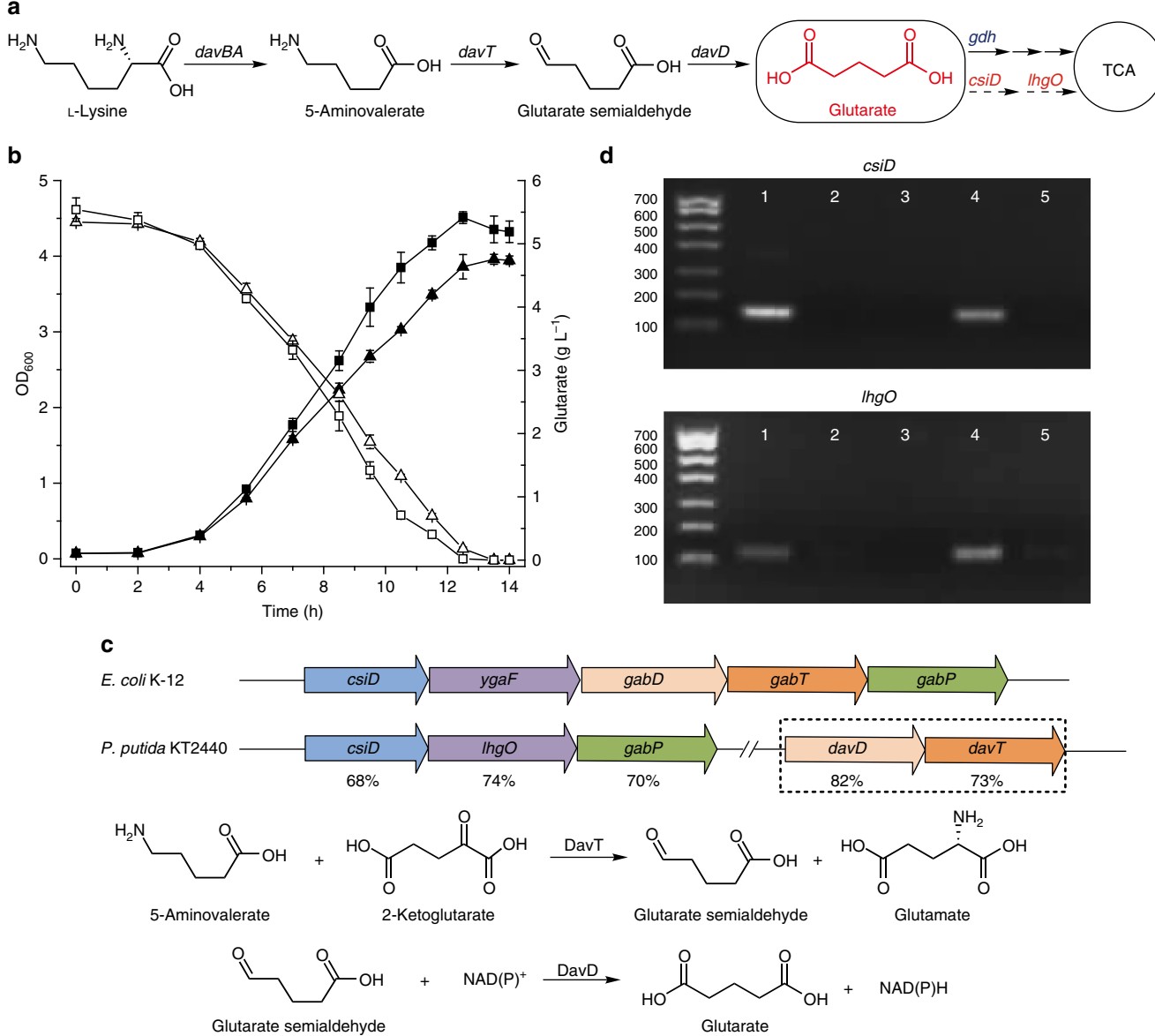

**Fig. 1** Glutarate hydroxylase and L-2-hydroxyglutarate oxidase are induced during glutarate utilization of *P. putida* KT2440. CsiD, glutarate hydroxylase; L-2-hydroxyglutarate, L-2-HG; LhgO, L-2-hydroxyglutarate oxidase. **a** The pathways of glutarate metabolism in *P. putida* KT2440. Besides the well-studied glutaryl-CoA dehydrogenation pathway requiring GDH, a catabolic pathway involved with L-2-HG anabolism and catabolism is proposed (in red). **b** Growth of *P. putida* KT2440 and its *gdh* mutant on glutarate. Growth (closed symbols) and the consumption of glutarate (open symbols) of wild-type *P. putida* KT2440 (squares) and its *gdh* mutant (triangles) was measured in MSM supplemented with 5 g L$^{-1}$ glutarate as the sole carbon source. Data shown are mean ± s.d. ($n = 3$ independent experiments). **c** Schematic representation of the gene clusters of *E. coli* K-12 and *P. putida* KT2440 containing *csiD* and *lhgO*. Orthologs are shown with matching colors and the identities of protein sequences are shown below the corresponding genes of *P. putida* KT2440. Arrows indicate the direction of gene transcription. The location of *davT* and *davD* in the genome of *P. putida* KT2440 and the roles of DavT and DavD involved in the glutarate production from 5-aminovalerate are also shown. **d** Agarose gel electrophoresis of *csiD* and *lhgO* RT-PCR products. RT-PCRs from mRNAs of *P. putida* KT2440 cells grown in 5 g L$^{-1}$ glutarate (lanes 2 and 4) or 5 g L$^{-1}$ pyruvate (lanes 3 and 5) as sole carbon sources were performed. The reactions were conducted in the presence of a reverse transcriptase (lanes 4 and 5) or in the absence of the enzyme (lanes 2 and 3, as a negative control). Genomic DNA was used as a positive control (lane 1). Lane M, molecular size marker. Numbers on the left present the sizes of the markers (in base pairs)

maintaining Fe$^{2+}$ in the reduced state[42,43]. The activity of CsiD was reduced to 42.6% in the absence of ascorbate (Fig. 2f). Additionally, the activity of CsiD was reduced to 80.3% in the absence of Fe$^{2+}$, suggesting that CsiD requires Fe$^{2+}$ for enzymatic activity as other standard Fe$^{2+}$/2-KG-dependent dioxygenases[44].

**CsiD is a glutarate hydroxylase producing L-2-HG.** The Fe$^{2+}$/2-KG-dependent dioxygenases couple the oxidative

decarboxylation of 2-KG to the oxidation (mostly hydroxylation) of the substrate[45,46]. In order to test whether CsiD can catalyze the hydroxylation of glutarate, the product of CsiD-catalyzed glutarate in the presence of 2-KG was analyzed by LC–MS/MS. As shown in Fig. 2g, when active CsiD was added to a reaction mixture containing 2-KG and glutarate, two compounds were produced, with elution times of 15.3 and 17.6 min. The molecular ion (M-H, m/z 147.0278) signal of the compound eluted at 15.3 min was consistent with 2-hydroxyglutarate (2-HG) (M.W.:

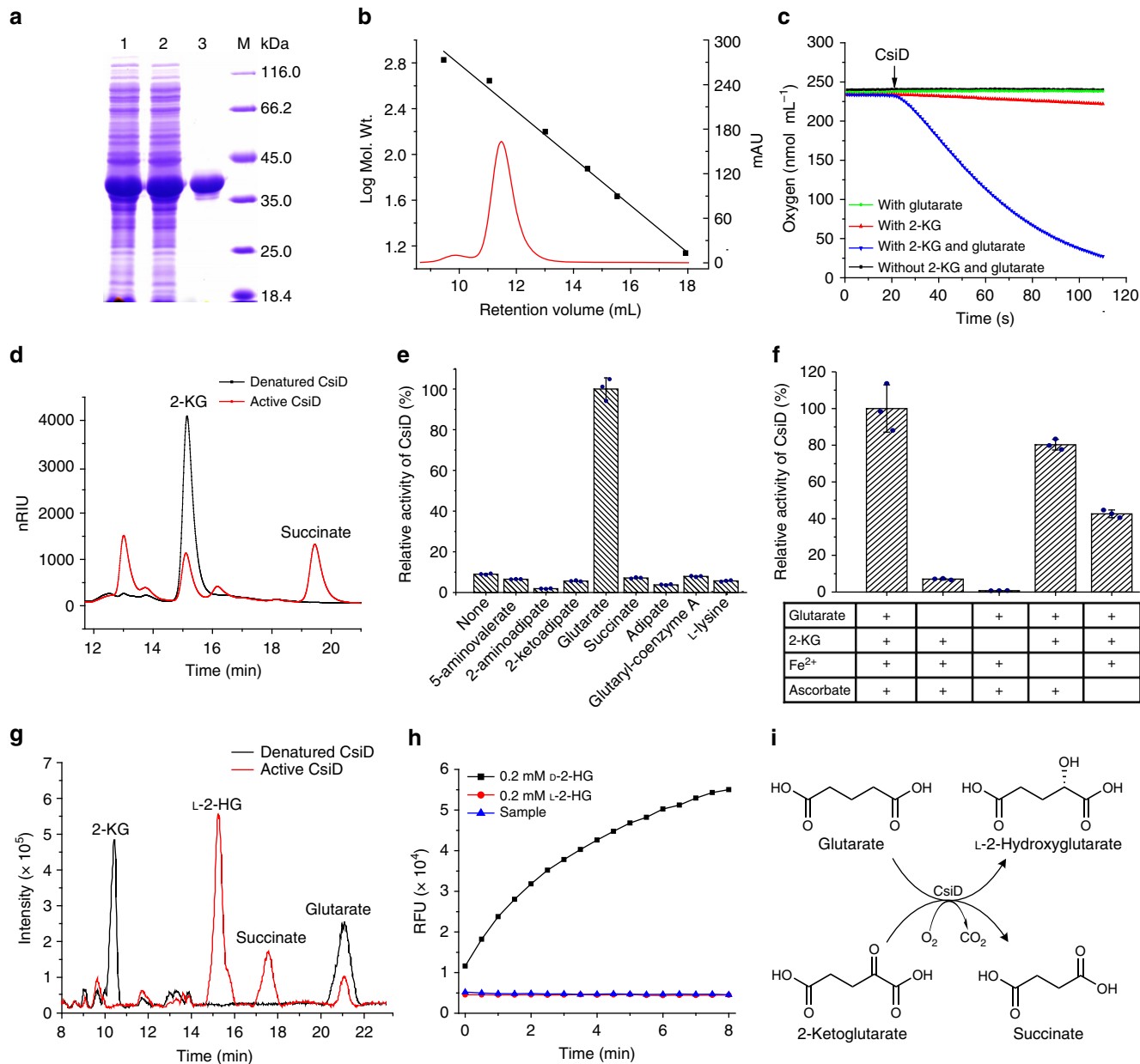

**Fig. 2** Purification and characterization of CsiD. **a** SDS–PAGE of expression and purification steps of CsiD. Lane M, molecular weight markers; lane 1, crude extract of *E. coli* BL21(DE3) harboring pETDuet-*csiD*; lane 2, the unbound protein of the HisTrap HP column; lane 3, purified CsiD using a HisTrap column. **b** The native molecular weight of CsiD determined by Superdex 200 10/300 GL. **c** The oxygen consumption of CsiD reacting with 2-KG in absence of glutarate (With 2-KG), with glutarate in absence of 2-KG (With glutarate), or with 2-KG and glutarate. **d** Analysis of the product of CsiD-catalyzed reaction toward 2-KG. Black line, the reaction with denatured CsiD; red line, the reaction with active CsiD; nRIU, nano-refractive index units. **e** Identification of the primary substrate of CsiD in *P. putida* KT2440 using an oxygen electrode. Data shown are mean ± s.d. ($n = 3$ independent experiments). **f** The components required for the catalytic activity of CsiD from *P. putida* KT2440. Data shown are mean ± s.d. ($n = 3$ independent experiments). **g** The total ion current chromatograms obtained by LC-MS/MS illustrating the product-forming behavior of CsiD-catalyzed 2-KG and glutarate. Black line, the reaction with denatured CsiD; red line, reaction mixture with active CsiD. **h** Chiral analysis of CsiD-catalyzed gluturate hydroxylation product by the enzymatic activity of HGDH. RFU, relative fluorescence units; Sample, CsiD-catalyzed product. **i** The schematic reaction catalyzed by CsiD

148.11) (Supplementary Fig. 1b).The molecular ion (M-H, m/z 117.0175) signal of the compound eluted at 17.6 min was consistent with succinate (M.W.: 118.09).

2-HG exists in two stereoisomeric forms: L-2-HG and D-2-HG[2,4]. As shown in Fig. 2h, D-2-HG was efficiently catalyzed by D-2-HG dehydrogenase (HGDH) from the anaerobic bacterium *Acidaminococcus fermentans*[47], whereas L-2-HG was not under the same assay conditions. When reaction products of CsiD were added to the assay mixture, no HGDH activity was detected

(Fig. 2h), indicating that the product obtained from glutarate by CsiD is L-2-HG, instead of D-2-HG.

The kinetic parameters of purified CsiD from *P. putida* KT2440 toward glutarate, 2-KG and oxygen were determined. The estimated $K_m$ values of CsiD for glutarate, 2-KG and oxygen were 145.67 ± 1.53 μM, 95.33 ± 0.97 μM, and 266.33 ± 13.58 μM, respectively. The estimated $V_{max}$ values of CsiD for glutarate, 2-KG and oxygen were 263.68 ± 2.32 U mg$^{-1}$, 144.10 ± 0.95 U mg$^{-1}$ and 272.33 ± 6.24 U mg$^{-1}$, respectively (Table 1).

**Table 1 Kinetic parameters of purified CsiD from *P. putida* KT2440[a]**

| Substrate | $K_m$ (µM) | $V_{max}$ (U mg$^{-1}$) | $K_{cat}$ (s$^{-1}$) | $K_{cat}/K_m$ (s$^{-1}$ µM$^{-1}$) |
|---|---|---|---|---|
| Glutarate | 145.67 ± 1.53 | 263.68 ± 2.32 | 1009.59 ± 11.50 | 6.93 ± 0.03 |
| 2-KG | 95.33 ± 0.97 | 144.10 ± 0.95 | 714.24 ± 4.73 | 7.49 ± 0.12 |
| Oxygen | 266.33 ± 13.58 | 272.33 ± 6.24 | 1349.83 ± 30.90 | 5.07 ± 0.15 |

[a]Data shown are mean ± s.d. (n = 3 independent experiments).

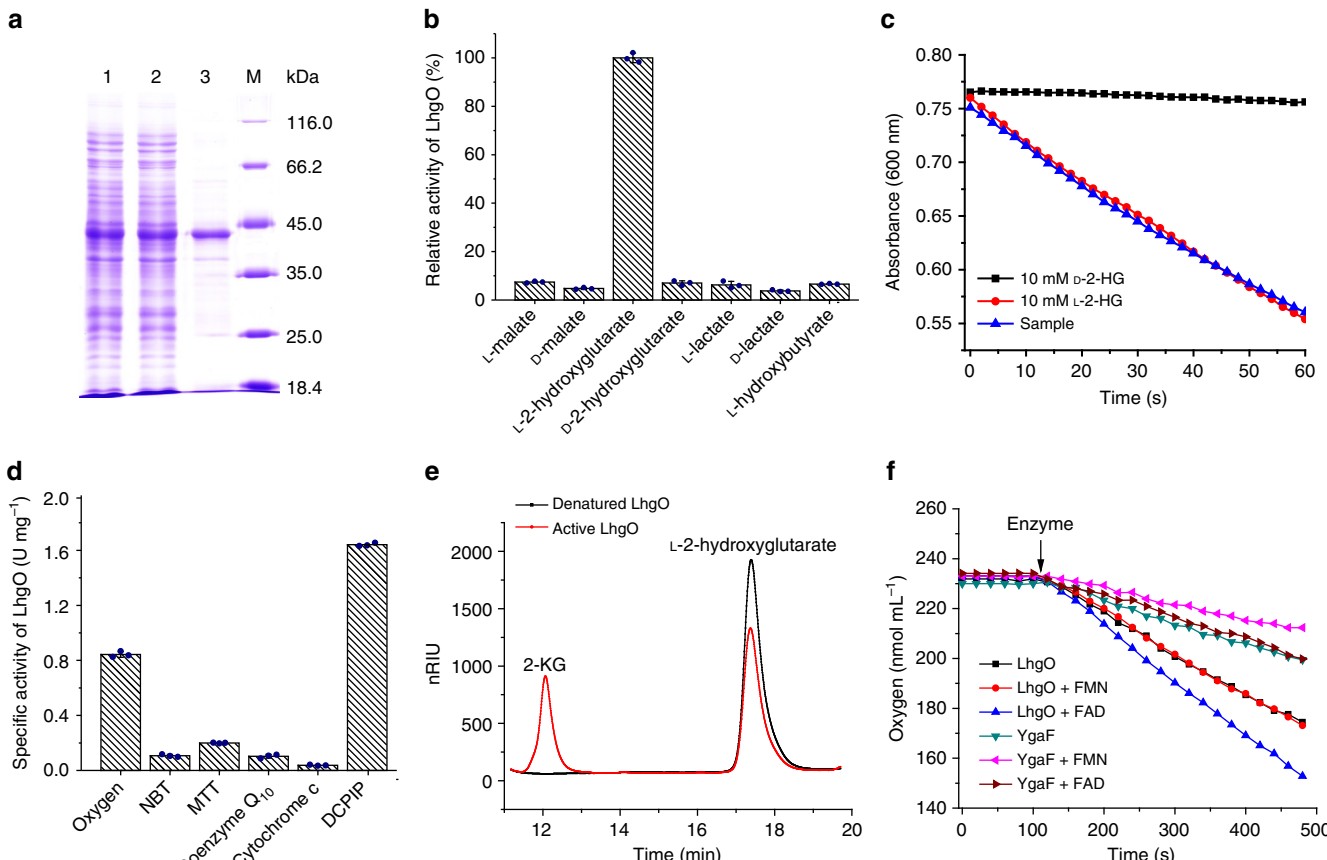

**Fig. 3** Purification and characterization of LhgO. **a** SDS–PAGE of expression and purification steps of LhgO. Lane M, molecular weight markers; lane 1, crude extract of *E. coli* BL21(DE3) harboring pETDuet-*lhgO*; lane 2, the unbound protein of the HisTrap HP column; lane 3, purified LhgO using a HisTrap column. **b** Substrate specificity of LhgO in *P. putida* KT2440. Data shown are mean ± s.d. (n = 3 independent experiments). **c** The time course of reactions of LhgO with L-2-HG, D-2-HG and CsiD-catalyzed product. Sample, CsiD-catalyzed product. **d** Specific activities of LhgO with different electron acceptors. Data shown are mean ± s.d. (n = 3 independent experiments). **e** HPLC chromatograms illustrating the product-forming behavior of LhgO in *P. putida*. Black line, the reaction mixture with denatured LhgO; red line, the reaction mixture with active LhgO. nRIU, nano-refractive index units. **f** The activities of LhgO and YgaF with molecular oxygen and the effects of FAD and FMN. LhgO and YgaF (1 mg mL$^{-1}$) were incubated with FAD or FMN (LhgO + FAD, LhgO + FMN, YgaF + FAD, YgaF + FMN) for 1 h just before measurement. The assay mixture contained 50 mM Tris-HCl (pH 7.4) and 5 mM L-2-HG at 30 °C at 900 rpm. After monitoring the background for about 2 min, the enzyme was added, and the oxygen traces were monitored

Based on the results mentioned above, CsiD from *P. putida* KT2440 is a Fe$^{2+}$/2-KG-dependent dioxygenase acting on glutarate and producing L-2-HG and succinate as its end products. Thus, *csiD* encodes a glutarate hydroxylase in *P. putida* KT2440, which might represent a previously uncharacterized step in the utilization of glutarate (Fig. 2i).

**LhgO is an L-2-HG oxidase**. The *lhgO* gene in *P. putida* KT2440 is located immediately downstream of the *csiD* gene (Fig. 1c) and is annotated as an L-2-HG oxidase. The *lhgO* gene was cloned and overexpressed in *E. coli* BL21(DE3), and the protein with a His tag was purified by affinity chromatography (Fig. 3a). The substrate specificity of LhgO was determined with 5 mM isomers of 2-hydroxy acids with different side chains using dichlorophenol-indophenol (DCPIP) as the electron acceptor. The LhgO seemed to have high substrate specificity. Robust activity was detected only when L-2-HG was used as the substrate (Fig. 3b). The activity of LhgO toward L-2-HG, D-2-HG, or the CsiD-catalyzed glutarate hydroxylation product was also visualized by monitoring the absorbance of DCPIP at 600 nm. As shown in Fig. 3c, a rapid absorbance change was observed when LhgO was incubated with 10 mM L-2-HG, whereas no absorbance change was detected when L-2-HG was replaced with D-2-HG. Additionally, significant enzymatic activity of LhgO was detected with the CsiD-catalyzed glutarate hydroxylation product, further

supporting that CsiD catalyzes the glutarate hydroxylation to produce L-2-HG.

With regard to different electron acceptors, LhgO preferred to use DCPIP and oxygen as its electron acceptors. Besides DCPIP and oxygen, LhgO could also use MTT [3-(4,5-dimethylthiazol-2-yl)-2,5-diphenyltetrazolium bromide], NBT (nitro blue tetrazolium), coenzyme $Q_{10}$, and cytochrome $c$ as electron acceptors (Fig. 3d). The product of LhgO catalyzed dehydrogenation of L-2-HG was also investigated. The results showed that 2-KG was obtained from the reaction solution containing active LhgO, while no 2-KG was found in the reaction solution with denatured LhgO (Fig. 3e). The production of 2-KG was further confirmed by LC–MS/MS, with the molecular ion at 145.0124 Da (M-H), matching 2-KG (M.W.: 146.11) (Supplementary Fig. 2).

The abilities of LhgO and *E. coli* YgaF (an identified L-2-HG oxidase[41]) to use oxygen as a direct electron acceptor were assessed using a Clark-type oxygen electrode. The oxidase activity of LhgO can be stimulated by FAD, indicating that part of FAD cofactor was lost during the purification of LhgO. In addition, the oxygen consumption rate of LhgO was faster than that of YgaF at the same protein concentration (Fig. 3f). Thus, like YgaF from *E. coli* K-12, LhgO from *P. putida* KT2440 is a FAD-dependent L-2-HG oxidase.

**GDH and CsiD support glutarate dependent growth.** As shown in Fig. 4a, *P. putida* KT2440 (Δ*csiD*) displayed a significantly delayed growth and a longer lag phase in glutarate medium than the wild-type strain. The phenotype of *P. putida* KT2440 (Δ*csiD*Δ*lhgO*) was identical with that of *P. putida* KT2440

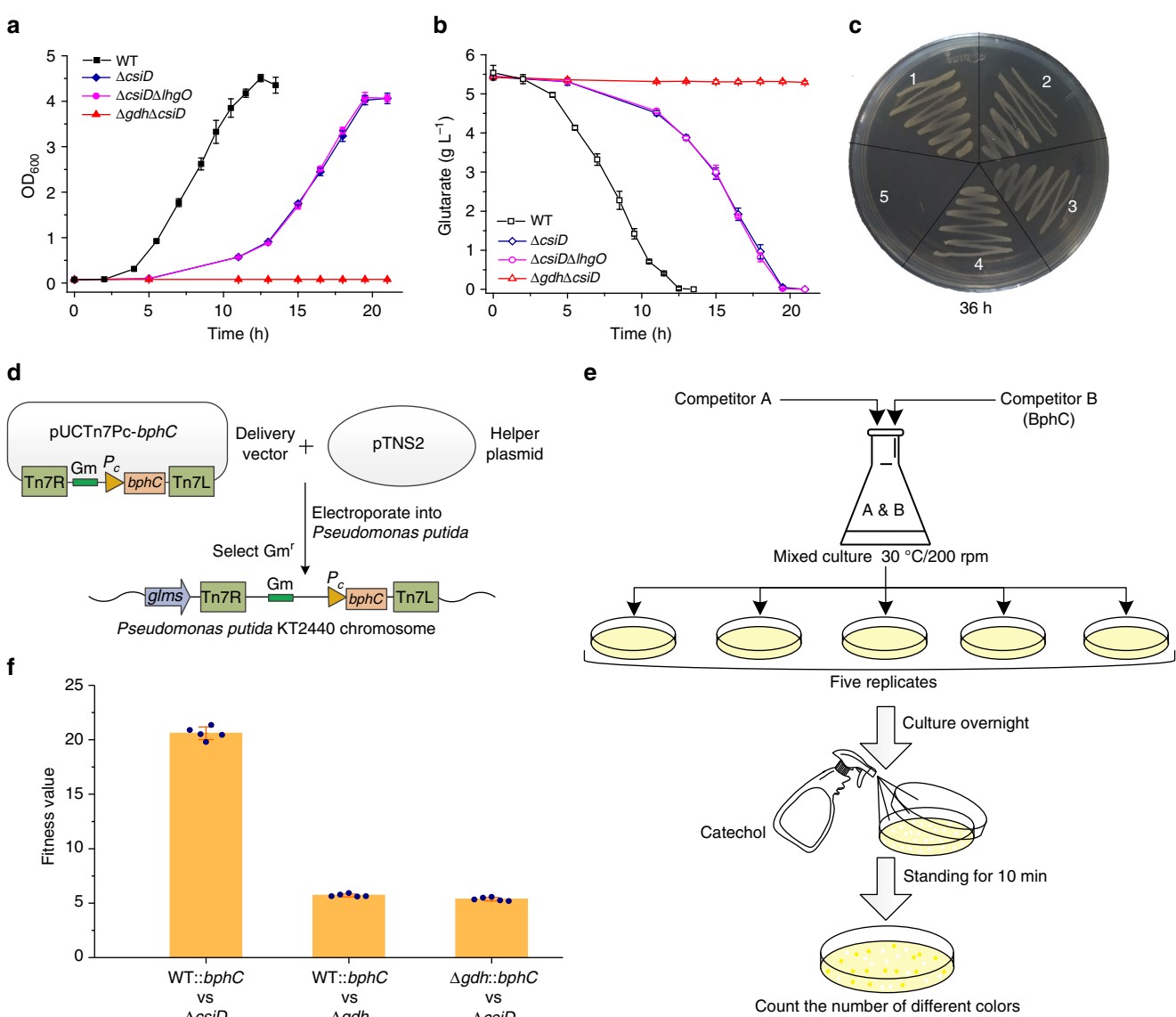

**Fig. 4** GDH and CsiD support glutarate dependent growth and the competitive fitness assays. **a** Growth of *P. putida* KT2440 and its glutarate hydroxylation pathway deletion mutants on glutarate. **b** The consumption of glutarate of wild-type *P. putida* KT2440 and its glutarate hydroxylation pathway deletion mutants on glutarate. **c** Growth of *P. putida* KT2440 and its derivatives on solid MSM containing 5 g L$^{-1}$ glutarate as the sole carbon source after 36 h. 1. *P. putida* KT2440; 2. *P. putida* KT2440 (Δ*csiD*); 3. *P. putida* KT2440 (Δ*csiD*Δ*lhgO*); 4. *P. putida* KT2440 (Δ*gdh*); 5. *P. putida* KT2440 (Δ*gdh*Δ*csiD*). **d** The construction scheme for producing *P. putida* KT2440 strains marked with *bphC*. **e** A schematic diagram of a competitive fitness assay. **f** Relative fitness of the glutaryl-CoA dehydrogenation pathway and the glutarate hydroxylation pathway in glutarate media. Fitness values (*W*) depict the fitness of Competitor 1 relative to Competitor 2. *W* = 1.0 indicates no difference in fitness between the two competitors. Values greater than 1 indicate a higher fitness of the first competitor relative to the second. Data shown are mean ± s.d. (*n* = 5 independent experiments)

($\Delta csiD$). As expected, *P. putida* KT2440 ($\Delta gdh\Delta csiD$) was incapable of using glutarate as its sole carbon source (Fig. 4a). The consumption of glutarate was correlated with the rates of growth (Fig. 4b). The wild-type *P. putida* KT2440 and its mutants were also cultured on solid minimal salt medium (MSM) containing 5 g L$^{-1}$ glutarate as the sole carbon source. *P. putida* KT2440 wild-type, *P. putida* KT2440 ($\Delta gdh$), *P. putida* KT2440 ($\Delta csiD$) and *P. putida* KT2440 ($\Delta csiD\Delta lhgO$) grew well, whereas *P. putida* KT2440 ($\Delta gdh\Delta csiD$) did not display any observable growth in 36 h (Fig. 4c). These data demonstrate that both GDH and CsiD participate in the degradation of glutarate and that both are functional in vivo. CsiD and LhgO constitute a glutarate hydroxylation pathway for glutarate utilization.

**Glutarate hydroxylation pathway has a competitive advantage**. There are two separate pathways (glutaryl-CoA dehydrogenation and glutarate hydroxylation) for glutarate metabolism in *P. putida* KT2440. The competitive fitness assays were conducted using *P. putida* KT2440 and the mutants with only the glutaryl-CoA dehydrogenation pathway or the glutarate hydroxylation pathway. A mini-Tn7 system was used to mark one of the competitors with a functional copy of *bphC*, encoding a dihydroxybiphenyl dioxygenase (BphC)[48,49] (Fig. 4d). The schematic diagram of the competitive fitness assays is shown in Fig. 4e. The ratio of the two competitive strains was close to 1:1 at the beginning of cultivation in MSM containing glutarate as the sole carbon source. After incubation for 24 h, the relative fitness was calculated from the ratio of bacterial colonies with different color reactions after catechol addition (Supplementary Fig. 3).

As shown in Fig. 4f, the wild-type strain showed a distinct fitness advantage toward the *P. putida* KT2440 ($\Delta csiD$) harboring the glutaryl-CoA dehydrogenation pathway ($W = 20.64 \pm 0.65$) and *P. putida* KT2440 ($\Delta gdh$) harboring the glutarate hydroxylation pathway ($W = 5.70 \pm 0.19$). In addition, *P. putida* KT2440 ($\Delta gdh$) harboring the glutarate hydroxylation pathway was more competitive than *P. putida* KT2440 ($\Delta csiD$) harboring the glutaryl-CoA dehydrogenation pathway ($W = 5.35 \pm 0.16$). The growth rate during exponential growth period and the maximum specific growth rate of *P. putida* KT2440 ($\Delta gdh$) are also both higher than that of *P. putida* KT2440 ($\Delta csiD$) (Supplementary Table 1), demonstrating that the glutarate hydroxylation pathway provides a competitive advantage over the glutaryl-CoA dehydrogenation pathway during glutarate utilization.

The expression of *csiD* and *gdh* during growth in the glutarate medium was analyzed by RT-PCR and quantitative real-time PCR (qPCR). Both *csiD* and *gdh* were induced at the beginning of growth in *P. putida* KT2440. *csiD* in *P. putida* KT2440 ($\Delta gdh$) and *gdh* in *P. putida* KT2440 ($\Delta csiD$) were also immediately expressed at the beginning of the growth (Supplementary Figs. 4 and 5). Additionally, *csiD* and *gdh* were both induced at the beginning of the utilization of L-lysine (Supplementary Fig. 6) and carbon starvation (Supplementary Fig. 7). We also complemented *csiD* or *gdh* in *P. putida* KT2440 ($\Delta gdh\Delta csiD$). The expression of *csiD* or *gdh* was controlled by $P_c$ (a constitutive promoter) or $P_{tac}$ (an inducible promoter). The *csiD* complement strain displayed a faster growth rate than *gdh* complement strain under the same promoter (Supplementary Fig. 8). Therefore, the competitive advantage of the glutarate hydroxylation pathway toward the glutaryl-CoA dehydrogenation pathway was not caused by the order of each pathway to be induced.

**Increased glutarate production by blocking the two pathways**. Glutarate is an attractive C5 platform chemical[24] that can be produced from L-lysine (Fig. 5a). *P. putida* KT2440 ($\Delta gdh\Delta csiD$) displays negligible growth on 5 g L$^{-1}$ L-lysine as the sole source of carbon and nitrogen. The concentration of glutarate in this strain increased to 0.04 g L$^{-1}$ after cultivation for 24 h, whereas the wild-type *P. putida* KT2440, *P. putida* KT2440 ($\Delta gdh$), and *P. putida* KT2440 ($\Delta csiD$) could not accumulate any glutarate (Fig. 5b, c). When 5 g L$^{-1}$ glucose and 5 g L$^{-1}$ L-lysine were added, *P. putida* KT2440 ($\Delta gdh\Delta csiD$) produced 1.92 g L$^{-1}$ glutarate after cultivation for 24 h, and there was also no accumulation of glutarate in the wild-type *P. putida* KT2440 and the single gene mutant strains (Fig. 5d, e). We also constructed *P. putida* KT2440 ($\Delta gdh\Delta csiD\Delta alr$) by deletion of the alanine racemase gene (*alr*) in *P. putida* KT2440 ($\Delta gdh\Delta csiD$), which blocked the conversion of L-lysine to D-lysine[30]. *P. putida* KT2440 ($\Delta gdh\Delta csiD\Delta alr$) produced 1.94 g L$^{-1}$ glutarate from L-lysine with a higher molar conversion ratio of 0.85 (Fig. 5f).

**Phylogenetic analyses of CsiD, LhgO, and GDH**. The distribution of CsiD, LhgO, and GDH in bacteria was studied by using a BLASTP program in the sequenced bacterial genomes from GenBank (updated April 6, 2016). Homologs of CsiD were found in 454 *Proteobacteria*. Homologs of LhgO were found in 608 *Proteobacteria* and 4 *Actinobacteria* (Supplementary Table 2). We overexpressed, purified, and biochemically characterized the CsiD and LhgO homologs from *E. coli* K-12 MG1655, *Klebsiella pneumoniae* ATCC25955 and *Salmonella enterica serovar Typhimurium* CT18, respectively. All three homologs of CsiD can catalyze the hydroxylation of glutarate, and the homologs of LhgO can catalyze L-2-HG to produce 2-KG (Supplementary Table 3). Homologs of GDH are found in 1004 bacterial genomes, including 876 *Proteobacteria*, 100 *Bacteroidetes*, 14 *Actinobacteria*, and a few species of *Acidobacteria*, *Gemmatimonadetes*, *Deinococcus-Thermus*, and *Ignavibacteriae* (Fig. 6a and Supplementary Table 2). As for *Pseudomonas* species, GDH is present universally, while homologs of CsiD and LhgO are only found in 25 and 48 species, respectively. Different from the widely distributed GDH, the distributions of CsiD and LhgO are relatively narrow and sporadic in *Pseudomonas*, implying that they might be acquired via horizontal gene transfer in *Pseudomonas*. Among the 4929 completely sequenced bacteria, only 26 bacteria are found to contain CsiD, LhgO, and GDH simultaneously, including 25 species of *Pseudomonas*, and one *Halomonas* (Supplementary Table 4).

Phylogenetic analysis revealed that homologs of CsiD could be only found in bacteria (Fig. 6b), whereas homologs of LhgO are widespread in eukaryotic microorganisms, animals, and plants besides bacteria (Fig. 6c). In addition, the phylogenomic distribution of LhgO in bacteria is divided into two distinct groupings (Fig. 6a, c). In the first group (164 genomes), only LhgO is present. LhgO is likely to convert L-2-HG to 2-KG as a metabolite repair enzyme, which is similar to the situation in mammals[6,7,50]. In the second group (448 genomes), both CsiD and LhgO are present. Given the participation of LhgO in the glutarate hydroxylation pathway, LhgO should not be considered solely as a metabolite repair enzyme but also as an important enzyme to degrade organic compounds.

**Discussion**
The end product of the glutaryl-CoA dehydrogenation pathway is crotonyl-CoA, which can then be converted into two molecules of acetyl-CoA (Fig. 7a). CsiD is a glutarate hydroxylase that uses oxygen, 2-KG and glutarate as the substrates, and $CO_2$, succinate and L-2-HG are produced. LhgO is an L-2-HG oxidase using L-2-HG as its substrate, and producing 2-KG. Thus, the end product of the glutarate hydroxylation pathway is indeed succinate, a C4 compound (Fig. 7a). The glutaryl-CoA dehydrogenation pathway and glutarate hydroxylation pathway can supply the C2 (acetyl-

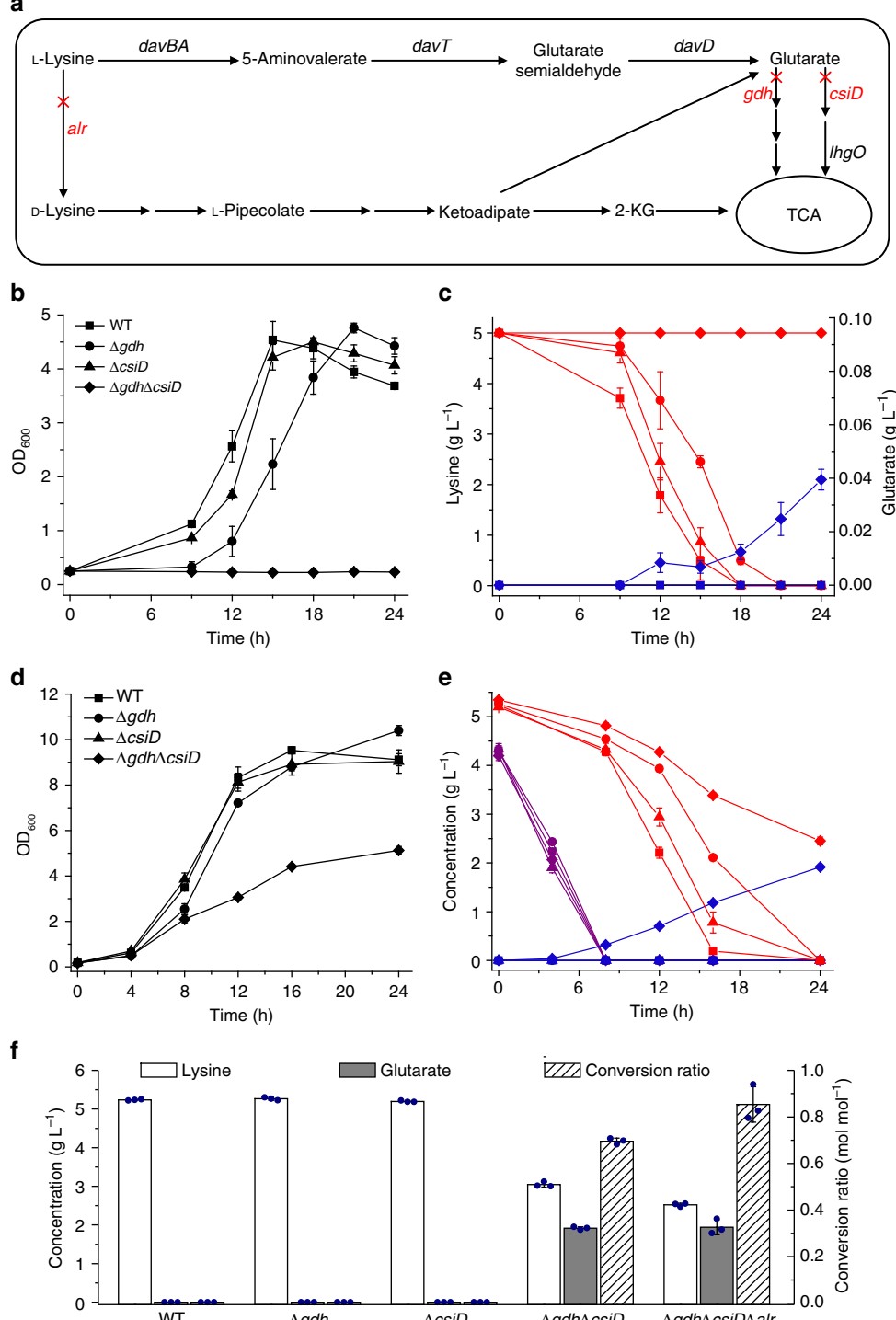

**Fig. 5** Production of glutarate in mutant of glutarate hydroxylation pathway and glutaryl-CoA dehydrogenation pathway. **a** The metabolic engineering strategies for the production of glutarate from L-lysine in *P. putida* KT2440. In this work, *gdh*, *csiD* and *alr* were inactivated individually or in combination for improvement of glutarate yield. *alr*, alanine racemase; *davBA*, L-lysine monooxygenase and 5-aminovaleramide amidohydrolase; *davT*, 5-aminovalerate aminotransferase; *davD*, glutaric semialdehyde dehydrogenase; 2-KG, 2-ketoglutarate. Growth (**b**) and the production of glutarate (**c**) by wild-type *P. putida* KT2440 and mutants cultured in L-lysine were compared. The consumption of L-lysine (red lines) and the yield of glutarate (blue lines) were shown. Growth (**d**) and the production of glutarate (**e**) by wild-type *P. putida* KT2440 and mutants cultured in the medium with L-lysine and glucose were compared. The consumption of L-lysine (red lines), the consumption of glucose (purple lines) and the yield of glutarate (blue lines) were shown. **f** Comparison of the consumption of L-lysine, the yield of glutarate and the conversion ratio using wild-type *P. putida* KT2440 and mutants cultured in the medium with L-lysine and glucose. Data shown are mean ± s.d. (*n* = 3 independent experiments)

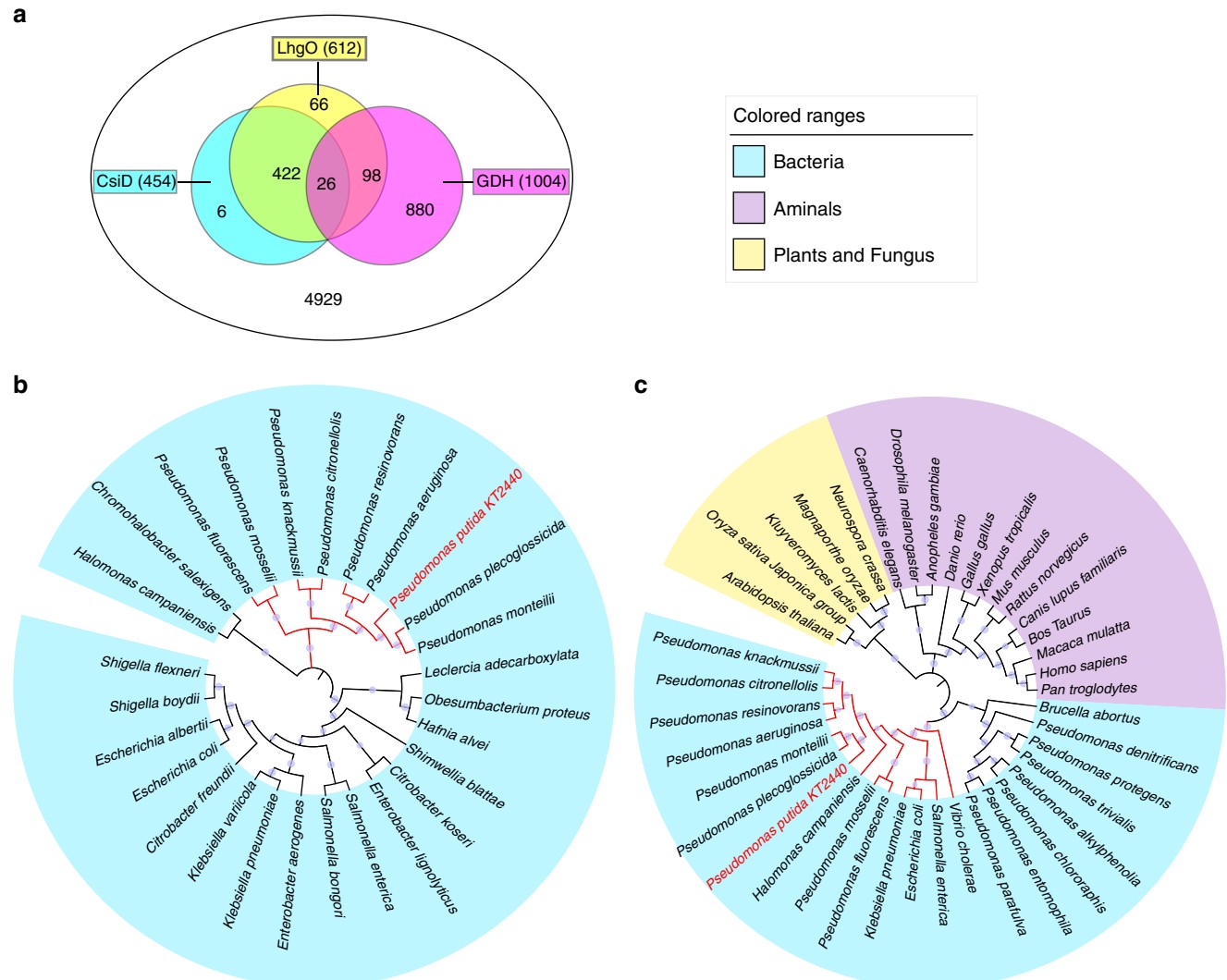

**Fig. 6** Phylogenetic distribution of CsiD and LhgO. **a** Distribution of CsiD, LhgO and GDH in bacteria. The Venn diagram illustrates the occurrence and overlap of CsiD, LhgO and GDH through genome context analysis. **b** Phylogenetic analysis of CsiD. Proteins are listed in Supplementary Table 5, with the organism origin and accession number. **c** Phylogenetic analysis of LhgO. Proteins are listed in Supplementary Table 6, with the organism origin and accession number

CoA) and C4 (succinate) compounds to the TCA cycle, respectively. *P. putida* KT2440 showed a distinct fitness advantage over *P. putida* KT2440 (Δ*csiD*) and *P. putida* KT2440 (Δ*gdh*), indicating that the glutarate hydroxylation and glutaryl-CoA dehydrogenation pathways cooperate during glutarate metabolism. Furthermore, the glutarate hydroxylation pathway has a competitive advantage toward glutaryl-CoA dehydrogenation pathway (Fig. 4f and Supplementary Fig. 9). Time series gene expression analysis indicated that both *csiD* and *gdh* were induced at the beginning of growth using glutarate as the sole carbon source (Supplementary Figs. 4 and 5). When expressed under the same promoter, *csiD* could support faster growth of *P. putida* KT2440 (Δ*gdh*Δ*csiD*) than that of *gdh* (Supplementary Fig. 8). Succinate, the end product of glutarate hydroxylation pathway, is an intermediate of TCA cycle and can be used more quickly than crotonyl-CoA, the intermediate of glutaryl-CoA dehydrogenation pathway. The competitive advantage of glutarate hydroxylation pathway is not caused by its immediate expression but possibly due to its ability to provide the quickly utilizable metabolic intermediate, succinate. This notion might be further addressed by tracing experiment using $^{13}$C-labeled glutarate.

Two molecules of acetyl-CoA can also be condensed to one C4 compound via the glyoxylate cycle[51]. The isocitrate lyase encoded by *aceA* and malate synthase encoded by *glcB* are the key enzymes of glyoxylate cycle and are essential for acetate and fatty acid metabolism (Supplementary Fig. 10). The growth of *P. putida* KT2440 (Δ*aceA*) and *P. putida* KT2440 (Δ*glcB*) with glutarate was slightly lower than that of the wild-type strain, implying that the glyoxylate cycle also supplies a small part of the C4 compound needed for growth (Fig. 7b). *P. putida* KT2440 (Δ*csiD*-Δ*aceA*) and *P. putida* KT2440 (Δ*csiD*Δ*glcB*) could not grow on glutarate, demonstrating the necessity of the glyoxylate cycle during the utilization of glutarate in strains harboring only the glutaryl-CoA dehydrogenation pathway (Fig. 7b).

Given this information, we proposed a model for glutarate catabolism in *P. putida* KT2440 (Fig. 7c). Specifically, glutaryl-CoA is converted to crotonyl-CoA by GDH, which is then converted into two molecules of acetyl-CoA to supply a C2 compound to the TCA cycle. Glutarate can also be converted into succinate by CsiD and LhgO to supply a C4 compound to the TCA cycle. Lastly, the glyoxylate cycle, which can condense acetyl-CoA into a C4 compound, also participates in the glutarate

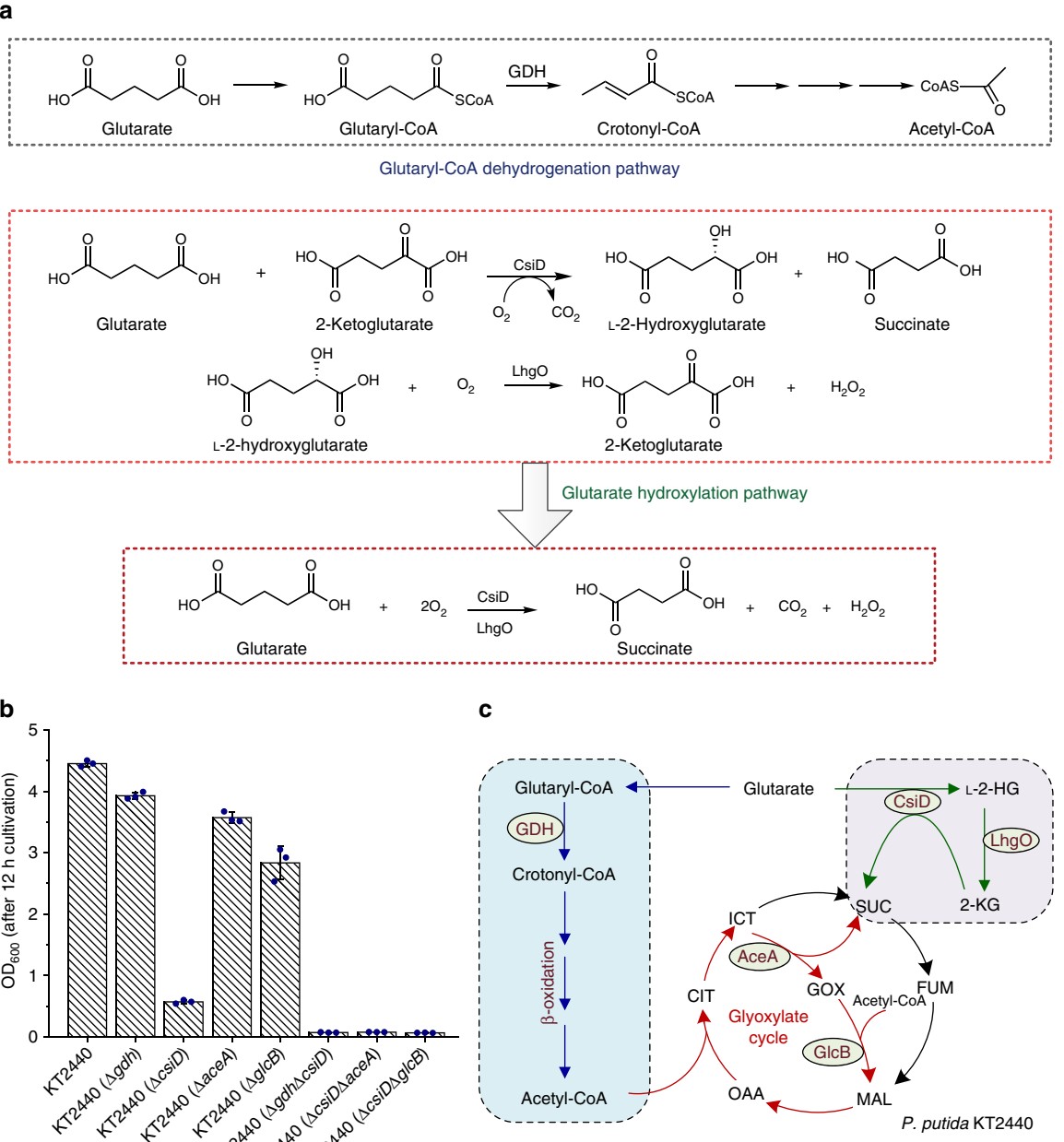

**Fig. 7** Metabolic pathways of glutarate in *P. putida* KT2440. **a** Analyses of the end products of the glutaryl-CoA dehydrogenation pathway and the glutarate hydroxylation pathway. **b** Comparison of the growth of *P. putida* KT2440 and its derivatives on glutarate. The biomass of *P. putida* KT2440 and its derivatives were estimated as the optical density at 600 nm after 12 h cultivation. Data shown are mean ± s.d. ($n = 3$ independent experiments). **c** Schematic representation of pathways for glutarate metabolism in *P. putida* KT2440. Blue-colored arrows indicate the glutaryl-CoA dehydrogenation pathway and green-colored arrows indicate the glutarate hydroxylation pathway. The acetyl-CoA generated from the glutaryl-CoA dehydrogenation pathway can be condensed to C4 compound via the glyoxylate cycle (red-colored arrows). AceA, isocitrate lyase; GlcB, malate synthase. GOX, glyoxylate; CIT, citrate; ICT, isocitrate; SUC, succinate; FUM, fumarate; MAL, malate; OAA, oxaloacetate; 2-KG, 2-ketoglutarate; L-2-HG, L-2-hydroxyglutarate

metabolism. These three pathways (i.e., the glutarate hydroxyla-tion pathway, glutaryl-CoA dehydrogenation pathway, and the glyoxylate cycle) cooperate to support efficient growth on gluta-rate (Fig. 7c).

Hypoxia can induce L-2-HG production in mammalian cells and increased intracellular L-2-HG is accompanied by the increased ratio of NADH to NAD$^+$[9,10]. Ratio of NADH to NAD$^+$ in *P. putida* KT2440 would increase with the decrease of rotational speeds (Supplementary Fig. 11a), but generation of L-2-HG decreased with the decrease of rotational speeds (Supplementary Fig. 11b, c). The expression and activities of CsiD

and LhgO in *P. putida* KT2440 also decreased with the decrease of the rotational speeds (Supplementary Fig. 12). This is different from the situation in mammalian cells but understandable since CsiD is a Fe$^{2+}$/2-KG-dependent dioxygenase requiring oxygen to produce L-2-HG and the presence of oxygen supporting L-2-HG production will decrease the ratio of NADH to NAD$^+$.

2-HG exists in two stereoisomeric conformations: L-2-HG and D-2-HG, which inhibit 2-KG-dependent enzymes involved in diverse biologic processes[1]. It is generally accepted that L-2-HG and D-2-HG are the reduction products of 2-KG. Mutations in isocitrate dehydrogenase, which result in strong D-2-HG

producing activity, have been detected in numerous types of cancer[3,52]. Although D-2-HG is viewed as a key oncogenic effector derived from isocitrate dehydrogenase mutations, we recently found that D-2-HG is a "hidden" but important metabolite produced by D-3-phosphoglycerate dehydrogenase in bacteria. The coupling between D-3-phosphoglycerate dehydrogenase and D-2-HG dehydrogenase drives L-serine biosynthesis, a crucial process for most bacteria[5]. It appears that L-2-HG is formed from the reduction of 2-KG by the promiscuous catalytic activity of L-malate dehydrogenase and L-lactate dehydrogenase in human cells under acidic and hypoxic conditions[9–14]. However, we demonstrated here that L-2-HG could also be produced from the hydroxylation of glutarate and is an intermediate of glutarate catabolism. Thus, L-2-HG and D-2-HG metabolisms are far more complicated than the currently accepted concept. Besides the reported enzymes and effects of L-2-HG and D-2-HG accumulation, other unidentified enzymes and components involved in L-2-HG and D-2-HG metabolisms are still worth studying.

Glutarate is also an attractive odd-carbon dicarboxylic acid with versatile applications[24]. Indeed, the bio-based route for glutarate production is now gaining worldwide attention[25,27,53,54]. However, the low yield of glutarate might be the bottleneck restricting its biotechnological production. Using the P. putida KT2440 ($\Delta gdh\Delta csiD\Delta alr$) without either the glutarate hydroxylation pathway or the glutaryl-CoA dehydrogenation pathway, the concentration of glutarate was enhanced to 1.94 g L$^{-1}$. More importantly, the yield of glutarate from L-lysine can be increased to 0.85 mol glutarate/mol L-lysine. Although GDH is absent, the homologs of CsiD and LhgO, which are the key enzymes in glutarate hydroxylation pathway, are present in E. coli. That might be the reason for the low yield of glutarate by using recombinant E. coli. Inactivation of CsiD and LhgO in E. coli might be a useful strategy to enhance glutarate production through biotechnological routes.

Degradation of the 20 amino acids leads to the formation of various ketogenic or glucogenic products. Thus, these amino acids can be divided into ketogenic or glucogenic. L-Lysine and L-leucine are viewed as ketogenic amino acids in all domains of life[55]. Glutaryl-CoA dehydrogenation pathway involves ketogenic chemicals, acetoacetate-CoA, and acetyl-CoA. The glutarate hydroxylation pathway involves the glucogenic chemicals, succinate and 2-KG. Based on the results in this work, L-lysine should be viewed as a ketogenic and glucogenic amino acid in strains with both the glutaryl-CoA dehydrogenation pathway and the glutarate hydroxylation pathway. In strains with only the glutarate hydroxylation pathway such as E. coli, L-lysine should be only viewed as a glucogenic amino acid.

The expression of csiD is activated by RpoS and repressed by CsiR in E. coli[36,37,56]. CsiR is allosterically regulated by some small compound produced during carbon starvation or when cells enter into stationary phase. CsiD and YgaF were supposed to be involved in production or catabolism of this compound[37]. The ortholog protein of CsiR (PP2908) locates upstream of CsiD (PP2909) in genome of P. putida KT2440. The expression of csiD and lhgO in P. putida KT2440 was induced by carbon starvation or exogenously added glutarate but repressed under stationary phase when glutarate added in medium had been depleted. CsiD and LhgO were found to construct a hydroxylation pathway for catabolism of glutarate. Thus, the expression of csiD and lhgO may also be regulated by CsiR in P. putida KT2440. Glutarate, a compound that can be produced from lysine degradation during carbon starvation or exogenously added, might be the effector of CsiR. The molecular function of CsiR and its regulation mechanism deserve further investigation.

In summary, we have demonstrated that CsiD in P. putida KT2440 is a $Fe^{2+}$/2-KG-dependent dioxygenase with the ability to hydroxylate glutarate, an additional route for L-2-HG production. LhgO is an L-2-HG oxidase, capable of converting L-2-HG into 2-KG. This sequence of L-2-HG anabolism and catabolism constitutes a glutarate hydroxylation pathway in P. putida KT2440. The glutarate hydroxylation pathway can supply a C4 compound for the TCA cycle more efficiently, which confers a competitive advantage over the well-known glutaryl-CoA dehydrogenation pathway. Besides a metabolite that can help mitigate cellular reductive stress and play a physiological role in adaptation to hypoxia, L-2-HG is also an important metabolic intermediate in the catabolism of several organic compounds.

## Methods

**Bacterial strains and culture conditions.** The bacterial strains and plasmids used in this study are listed in Supplementary Data 1. E. coli strains were cultured in Luria–Bertani (LB) medium at 180 rpm and 37 °C. Unless specified, P. putida KT2440 and its derivatives were cultivated in MSM[57] supplemented with 5.0 g L$^{-1}$ glutarate as the sole carbon source (glutarate medium) at 200 rpm and 30 °C. Antibiotics were added to the media when necessary, at the following concentrations: kanamycin at 50 μg mL$^{-1}$, gentamicin at 30 μg mL$^{-1}$ and ampicillin at 100 μg mL$^{-1}$.

**Construction of P. putida KT2440 mutants.** To construct the P. putida KT2440 ($\Delta gdh$) mutant strain, the homologous arms upstream and downstream of the gdh gene were amplified using the primers gdh-uf/gdh-ur and gdh-df/gdh-dr (Supplementary Data 2). The upstream and downstream fragments were fused together via recombinant PCR by using the primers gdh-uf and gdh-dr, which contained BamHI and HindIII restriction enzyme sites, respectively. The generated fusion and pK18mobsacB[58], a mobilizable plasmid that does not replicate in P. putida, were digested with BamHI and HindIII, respectively, and were then linked by using T4 DNA ligase to form pK18mobsacB-$\Delta gdh$. The plasmid was transferred into P. putida KT2440 via electrotransformation. The single-crossover mutants with integration of the plasmid pK18mobsacB-$\Delta gdh$ into the chromosome were selected on LB plate supplemented with 50 μg mL$^{-1}$ kanamycin. The second crossover cells were screened from LB plates containing 10% (w/v) sucrose. All the constructed strains were validated by PCR and sequenced. The csiD, lhgO, aceA, glcB, and alr mutants of P. putida KT2440 were generated by using the same procedure.

**RT-PCR and quantitative real-time PCR.** For RT-PCR experiments, total RNA was extracted from P. putida KT2440 cells grown in MSM supplemented with the appropriate carbon sources using an RNAprep pure Cell/Bacteria Kit (Tiangen Biotech, China). DNA contamination was eliminated by RNase-free DNase I (Transgen, China) and the quality of RNA was checked by 1.5% agarose gel electrophoresis. The total cDNA was synthesized using Superscript II Reverse Transcriptase (Transgen, China) in 20 μl reverse transcription reactions. Samples were initially heated at 65 °C for 5 min, placed on ice for 2 min, incubated at 25 °C for 10 min and 42 °C for 30 min. The reaction was terminated by incubation at 70 °C for 15 min. Reverse transcription-PCR (RT-PCR) was performed with the corresponding oligonucleotides (Supplementary Data 2). Total RNA and genomic DNA of P. Putida KT2440 were used as negative and positive controls, respectively.

For the quantitative real-time PCR (qPCR) assay, the total RNA preparation was obtained from three independent cultures (three biological replicates). RNA was transformed into cDNA using Superscript II Reverse Transcriptase (Transgen, China). The qPCR analysis was performed using TransStart Top Green qPCR SuperMix (Transgen, China) on the LightCycler 480 (Roche). The primers used are listed in Supplementary Data 2. For absolute quantification of the copy numbers, standard curves were constructed for each amplicon by 10-fold serial dilutions of the recombinant plasmids harboring the same amplicons (Supplementary Fig. 13). Each reaction was performed in triplicate. Controls with no template and no reverse transcription were included for each reaction on the same plate.

**Expression and purification of recombinant CsiD and LhgO.** The genes encoding csiD and lhgO were amplified from genomic DNA of P. putida KT2440 through PCR and csiD-F/csiD-R and lhgO-F/lhgO-R as primers, respectively (Supplementary Data 2). The products of csiD and lhgO were digested with EcoRI and HindIII, and then ligated into expression vector pETDuet-1 to obtain plasmid pETDuet-csiD (P. putida) and pETDuet-lhgO (P. putida). The constructed expression plasmids were separately transformed into E. coli BL21(DE3) for CsiD and LhgO expression. The recombinant E. coli strains were cultured to an optical density at 600 nm (OD$_{600}$) of 0.5 to 0.6 and induced at 16 °C with 1 mM isopropyl-D-1-thiogalactopyranoside (IPTG) for 10 h. The cells were harvested and washed twice with buffer A (20 mM sodium phosphate and 500 mM sodium chloride, pH 7.4), and then resuspended in the same buffer containing 1 mM phenylmethane-sulphonyl fluoride (PMSF) and 10% glycerol (v/v). Cells were disrupted by ultrasonic (Sonics 500 W; 20 kHz) on ice and then the cell lysate was centrifuged at 16,000 × g for 20 min at 4 °C to remove cell debris. The resultant supernatant was

loaded onto a HisTrap HP column (5 mL) equilibrated with buffer A and eluted with buffer B (20 mM sodium phosphate, 500 mM imidazole, and 500 mM sodium chloride, pH 7.4). The purified enzymes were concentrated by ultrafiltration, exchanged buffer with a Superdex G-25 column and analyzed by sodium dodecyl sulfate-polyacrylamide gel electrophoresis (SDS–PAGE) with 12.5% poly-acrylamide gels. Protein concentrations were determined by the Bradford assays. Finally, CsiD was stored in 50 mM Tris-HCl buffer (pH 7.4) at −80 °C and LhgO was stored in buffer C (50 mM sodium phosphate, and 150 mM sodium chloride, pH 7.2) at 4 °C, respectively. The expression and purification procedures of the homologs of CsiD and LhgO from *E. coli* K-12 MG1655 (also called as YgaF) and *K. pneumoniae* ATCC25955 were the same as those of *P. putida* KT2440. The primes used in the expression were shown in Supplementary Data 2. The predicted genes encoding the CsiD and LhgO homologs from *S. typhimurium* CT18 were synthesized by General Biosystems, Inc. (Anhui, China) and ligated into restriction sites of BamHI/HindIII of plasmid pETDuet-1. The subsequent expression and purification procedures were the same as those of *P. putida* KT2440.

The native molecular weight of CsiD in *P. putida* KT2440 was determined with a gel filtration column (Superdex 200 10/300 GL) using thyroglobulin (669 kDa), ferritin (440 kDa), aldolase (158 kDa), conalbumin (75 kDa), ovalbumin (43 kDa), and RNase A (13.7 kDa) as standard proteins. The eluent buffer was buffer C and a flow rate of 0.5 mL min$^{-1}$ was maintained throughout. The submolecular structure of the enzyme was studied under denaturing conditions by SDS–PAGE.

**Enzymatic assays of CsiD and LhgO**. The activity of CsiD was measured in 500 µl reaction solution, which contained 20 mM imidazole (pH 6.7), 1 mM glutarate, 1 mM 2-KG, 0.4 mM ascorbate, 50 µM Fe$^{2+}$, and 2 µg of purified CsiD, using a Clark-type oxygen electrode (Oxytherm, Hansatech, United Kingdom) equipped with an automatically temperature-controlled electrode chamber at 900 rpm. One unit (U) of CsiD activity was defined as the amount that catalyzed the reduction of 1 µmol of oxygen per minute. The double-reciprocal plot method was used to estimate the kinetic parameters of CsiD toward glutarate, 2-KG and oxygen. The total activity of CsiD was definied 100% when all of the above reaction compo-nents existed, and the effect of every component on the activity of CsiD was assessed by removing one of them from the reaction mixture.

The activity of LhgO was assayed at 30 °C in 50 mM Tris-HCl buffer (pH 7.4) containing 5 mM L-2-HG, 0.1 mM DCPIP and 0.021 mg mL$^{-1}$ purified LhgO, unless otherwise stated. The rate of DCPIP reduction was determined by measuring the absorbance change at 600 nm using a UV/visible spectrophotometer (Ultrospec 2100 pro, Amersham Biosciences, USA). ε value of DCPIP is 22 cm$^{-1}$ mM$^{-1}$. One unit (U) of LhgO activity was defined as the amount that catalyzed the reduction of 1 µmol of DCPIP per minute. The reactivity of LhgO with molecular oxygen was analyzed in 500 µL reaction solution using Clark-type oxygen electrode. Just before measurement, 1 mg mL$^{-1}$ LhgO was incubated with 50 µM FAD or FMN for 1 h on ice. The initial assay mixture contained 50 mM Tris–HCl (pH 7.4) and 5 mM L-2-HG at 30 °C and 900 rpm. After monitoring the background for about 2 min, 25 µL incubated LhgO (with FAD or FMN) was added, and the rate of oxygen consumption was monitored. The measurement of reactivity of YgaF in *E. coli* with molecular oxygen was the same as that of LhgO.

*P. putida* KT2440 was grown to mid-log stage (OD$_{600}$= 2) in 50 mL glutarate medium at different rotational speeds. Cells were harvested by centrifugation (6000 × *g* for 10 min at 4 °C), washed and suspended in phosphate-buffered saline (PBS) supplemented with 1 mM phenylmethylsulfonyl fluoride (PMSF). The final OD$_{600}$ of the cells was 20 and the cells were disrupted by sonication with a Sonics sonicator (500 W; 20 KHz) in an ice bath. The cell debris were removed by centrifugation at 13,000 × *g* for 5 min at 4 °C and the resultant supernatants were used as the crude cell extracts to measure the activities of CsiD and LhgO. The protein concentration of the crude cell extracts was determined by the method of Bradford, with bovine serum albumin as a standard.

**Identification of catalytic products of CsiD and LhgO**. To determine the product of the CsiD-catalyzed oxidative reaction using 2-KG as the substate, the reaction solution containing 10 mM 2-KG, 2 mM ascorbate, 0.25 mM Fe$^{2+}$, 0.25 mg mL$^{-1}$ purified CsiD in 50 mM Tris-HCl (pH 7.4) was incubated aerobically at 30 °C and 180 rpm for 1 h. For the product of CsiD-catalyzed glutarate hydroxylation, 10 mM glutarate was added besides the above components. The product of LhgO catalyzed dehydrogenation of L-2-HG was investigated in the reaction solutions containing 50 mM Tris-HCl (pH 7.4), 5 mM L-2-HG, 0.11 mg mL$^{-1}$ purified LhgO, and 1 mM MTT at 30 °C and 180 rpm for 1 h. The mixture was boiled to terminate the reaction, centrifuged at 20,000 × *g* for 15 min, and then subjected to HPLC analysis using an Aminex HPX-87H column (Bio-Rad) and a refractive index detector[59]. The reaction with denatured protein was conducted under identical conditions as a control. The catalytic product of CsiD was also analyzed by liquid chromatography-tandem mass spectrometry (LC–MS/MS, impact HD; Bruker Daltonics) using 0.1% formic acid at a flow rate of 0.4 mL min$^{-1}$ with a HPLC system coupled by negative electrospray ionization.

**Competitive fitness assays**. To distinguish between competitor strains, a mini-Tn7 system was used to mark one of the competitors with a functional copy of *bphC*[48]. We amplified gene *bphC* using the vector pMMPc-*bphC*[60] as a template

through PCR with primer pairs *bphC*-F (SmaI)/*bphC*-R(SpeI) (Supplementary Data 2). Then, the amplified fragments were inserted into the SmaI and SpeI sites of pUCTn7Pc[60], generating pUCTn7Pc-*bphC*. The constructed vector and pTNS2[48] was transferred into *P. putida* KT2440 wild-type and *P. putida* KT2440 (Δ*gdh*) simultaneously by electroporation, respectively. The competition strains marked with *bphC* colored yellow when they were sprayed with 0.2 M catechol, whereas the unmarked strains not.

To analyze the competitive fitness between strains, the pre-activated competitors were mixed in the glutarate medium at a 1:1 volumetric ratio and cultivated for 24 h. Initial and final competitor ratios were determined by plating on LB agar plates (five technical replicates in each independent experiment) at 0 h and 24 h. Control experiments showed that strains marked with *bphC* had no effect on its fitness under the conditions used. Each relative fitness assay value is the mean of five biological replicates.

**Bioinformatics analysis**. The phylogenetic trees of CsiD and LhgO were con-structed from the alignment of multiple proteins (Supplementary Tables 5 and 6) using the program ClustalX version 2.1, followed by neighbor-joining analysis using the MEGA software program (version 5.10) with bootstrap analysis for 1000 replications. The phylogenetic trees were processed by Interactive tree of life (iTOL) v3[61]. The distributions of CsiD, LhgO and GDH in bacteria were vigorously checked by searching the sequenced bacterial genomes from GenBank (updated until April 6, 2016) with the GDH, CsiD and LhgO protein sequences as the queries. Individual genomes showing the query coverage of more than 90%, an E value lower than e$^{-30}$, and a maximum identity level higher than 50% with the query protein were selected and further evaluated.

Additional materials and methods could be found in the Supplementary Methods part of the Supplementary Information file.

**Data availability**. The data supporting the findings of this study are available within the article and its Supplementary Information files and from the corre-sponding authors on requst.

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

## Acknowledgements

We thank Jing Zhu and Zhifeng Li from Analysis & Testing Center of SKLMT (State Key laboratory of Microbial Technology, Shandong University) for assistance in liquid chromatography-tandem mass spectrometry (LC–MS/MS) of Ultimate3000-impact HD (Bruker) and quantitative real-time PCR of LightCycler 480 (Roche). This work was supported by the National Natural Science Foundation of China (31470199 and 31470164), Shandong Provincial Funds for Distinguished Young Scientists (JQ 201806), and the Young Scholars Program of Shandong University (2015WLJH25).

## Author contributions

M.Z. designed the research, performed most experiments, analyzed the data, and wrote the manuscript. C.G. designed the research, contributed the bioinformatics, analyzed the data, and wrote the paper. X.G. conducted biochemical assays and competitive fitness assays. S.G. contributed determination of intracellular NAD$^+$ and NADH concentrations. Z.K. performed carbon starvation experiments. D.X. and J.Y. prepared samples for

qPCR. W.Z. contributed bioinformatics. W.D. contributed the MS analysis. P.L. contributed quantification of L-lysine. F.T. and C.Y. analyzed the data and contributed discussion. C.M. and P.X. designed the research, analyzed the data, and wrote the paper.

## Additional information

**Competing interests:** The authors declare no competing interests.

