## [Peer Review File · Nature Communications]

Reviewers' comments:

Reviewer #1 (Remarks to the Author):

In this manuscript, the authors provide convincing molecular and biochemical evidence for a novel pathway of L-2-hydroxyglutarate synthesis in select prokaryotes.

--What is the effect of hypoxia on CsiD and LGO expression and activity, and on L2HG generation in *P. putida*?

--How does intracellular redox state modulate L2HG synthesis by this pathway (i.e., how do changes in GSH/GSSG alter synthesis)?

--The concluding sentence in the discussion, "L-2-HG should not only be treated as an abnormal metabolite, but should be recognized as an important metabolic intermediate during degradation of some organic compounds," is an appropriate conclusion based on the data, but hardly a novel concept (cf. your references 11 and 12).

Reviewer #2 (Remarks to the Author):

The authors have discovered a new pathway for the breakdown of L-lysine. The key enzyme of this pathway is CsiD. They show this by proving the GDH is not indispensable for glutarate utilization, that CsiD is induced during glutarate utilization, Knocking out both GDH and CsiD rendered the strain unable to grow on glutarate, They show CsiD is a Fe²⁺/2-KG-dependent deoxygenase producing L-2-HG.

The document shows a novel pathway, containing a new enzyme. The document is well written and deserves publication in Nature communications. A few things have to be changed though.

Major comments

line 111: why shift from *E. coli* to *P. putida*? Please explain.

The phylogenetic analysis should be part of the results section.

The phylogenetic analysis revealed many homologues, but that doesn't prove that they have the same function. Showing enzymatic activity is necessary to prove this. The scope of the paper should therefore be changed to *P. putida* only, general proof to allow "bacteria" as scope is not

delivered.

Please mention the EC number of the enzyme

Line 45-50, 62-65. The link to pathogenesis is too strong in both introduction and discussion section. It suggests the authors are looking for a cure.

Minor comments

Title: The authors have shown that the new pathway is active in *Pseudomonas putida* kt2442, not in other bacteria. Limit the scope of the title.

Line 33. Remove "then"

line 34: change "it can produce" into "it produced"

Line 35: "bacterial" has a too wide scope.

Line 73-76: not necessary, please remove

Line 84: "microbial" has a too wide a scope.

Line 107: reference 45 does not mention CsiD. Please explain how you come to the conclusion that CsiD is a non-hem FE₂⁺ etc....

line 295-303. Too speculative

Reviewer #3 (Remarks to the Author):

The manuscript present by Zhang and colleges is an interesting metabolic study which uncovers a new metabolic pathway for glutarate metabolism in *P. putida*. Specifically, they identify an Fe²⁺/2-KG-dependent glutarate hydroxylase, unknown so far, encoded by the gene *csiD*. *CsiD* transform glutarate into L-2-Hydroxyglutarate which is further converted in ketoglutarate driven by a new L-2-HG dehydrogenase encoded by the gene *lgo*. Overall, the new pathway yield succinate from glutarate and supposes an alternative pathway to the well-known glutaryl-coenzyme A (CoA) dehydrogenation pathway which yield 2 acetyl-CoA molecules. By knocking out both pathways in *P. putida*, they further construct a glutarate overproducer strain from lysine. They provide enough physiological, genetics, and biochemical experiment supporting these results.

Finally, they discuss the presence of both pathways in *P. putida* by means of metabolism optimization. Thus, since the new pathway produce a gluconeogenic final product such as succinate, and the dehydrogenation pathway acetyl-CoA feeding TCA, the simultaneous operation of both pathways should provide a metabolic advantage over strain harboring a single pathway for glutarate degradation. Such results, which constitute one of the more interesting aspects of this manuscript, are poorly supported by direct evidences. Therefore, alternative hypothesis cannot be left aside and the supposed advantage of having two pathways over single pathway should be strongly supported with additional experiments.

Major Comment

Along the manuscript glutarate hydroxylation pathway is presented having a competitive advantage toward the dehydrogenation pathway. However the nature of this advantage is not clearly demonstrated. *P. putida* has high functional redundancy including several ecoparalog enzymes and pathways. Examples include multiple isoenzymes for key TCA steps and multiple catabolic pathways for glucose and amino acids metabolisms. This fact has been explained always in term of metabolic robustness.

1. The authors clearly demonstrate that *csiD* is strongly induced in the presence of glutarate. What about the expression of genes encoding the dehydrogenase pathway? Are they induced?. According the large lag phase found in the *csiD* knockout, it could be concluded that genes encoding the dehydrogenase pathway are solely induced after a large carbon starving stress and not from the beginning. Ecoparalog or redundant pathways often are deferentially expressed responding to specific environmental conditions. It could be possible that under optimal conditions (those used in the study) only the genes from the hydroxylation pathway are strongly induced?. A time series gene expression analysis of genes from both pathways could clarify these aspects. Monitoring the gene expression of both pathway under different environmental stresses could be of interest. Also, what about lysine metabolism? Under these conditions (glutarate is produced in the cytoplasm) the behavior of *csiD* and *gdcH* knockouts is the same?. Please clarify.

2. Pg. 11. Fig. 4

"As shown in Fig. 4f, the wild-type strain show a distinct fitness advantage toward the *P. putida* KT2440 (Δ *csiD*) harboring the glutaryl-CoA dehydrogenation pathway ($W = 20.64$) and *P. putida* KT2440 (Δ *gdh*) harboring the glutarate hydroxylation pathway ($W = 5.70$). In addition, *P. putida* KT2440 (Δ *gdh*) harboring the glutarate hydroxylation pathway was more competitive than *P. putida* KT2440 (Δ *csiD*) harboring the glutaryl-CoA dehydrogenation pathway ($W = 5.35$), implying that the glutarate hydroxylation pathway provides a competitive advantage over the glutaryl-CoA dehydrogenation pathway during glutarate utilization."

The supposed advantage of hydroxylation pathway addressed by these competition experiments also is questionable. If the dehydrogenation genes are not induced from the beginning, is not possible to conclude whether the advantage is because a metabolic or regulation bottlenecks. Solving the gene expression profile of both pathways as discussed above will help in this topic. From the Fig. 4a, can be concluded that excluding the lag phase, the growth rate and final

biomass yield of *csiD* knockout is similar than that from the WT strain. Therefore, it's not evident this advantage in term of metabolic optimization but again these results argue in favor of regulatory bottleneck. In other words, the hydrogenation pathway seems as efficient as hydroxylation pathway after complete induction. Exhaustive and quantitative estimation of growth rates, final biomass yields etc for each knockout strain should be done and included in the paper to clarify, even more, the yield of each pathway.

3. Fig. 6. The authors discuss a very interesting synergy between both pathways. Thus, while the hydroxylation pathway would provide C4 carbon products, the dehydrogenation pathway would be in charge of providing Acetyl-CoA. Overall, the activation of both pathways would be an advantage over single pathways. In any case, the activation of dehydrogenation pathway has not been demonstrated. The experiments using knockout strains on glyoxylate cycle only provide indirect evidences since they are taken after 12 h of cultivation. This 12 h seems to be a magic number. Maximum OD is achieved by the wt and *gdch* knockout (Fig. 1). The lag phase in the *csiD* knockouts is around 12 h (Fig.4). 12 h are needed to start the consumption of glutarate in this same knockout strain (Fig.4). What is the behavior of such knockouts strain at early time?. There is not synergy at mid exponential phase?.

Experiment carried out with the *gdch* knockout (Fig. 1) could suggest that the role of dehydrogenation pathway is solely needed at the end of exponential phase. Again, the differential expression of both pathways is on the table.

The synergistic action of several redundant pathways operating in response of different environmental stress is very interesting and widely extended in robust bacteria such as *P. putida*. Much more direct evidence should be desirable here. ¹³C experiments could elucidate the role of each pathway in glutarate metabolism. In addition, the above indicate gene expression profiles could help.

4. This reviewer is fully convinced of the need to increase biochemical studies providing new enzymatic activities and metabolic pathways. The advent of genomics has overshadowed detailed biochemistry studies and, as a fatal consequence, new activities are now assigned, in many cases, based on bioinformatics evidences with low confidence. For this reason, I considered that the new activities found here have a great value. However, since the activity of *CsiD* has been reported here for the first time, I miss basic biochemical characterization of this interesting enzyme. The authors should consider providing this information. At least km for each substrate, V_{max} , k_{cat} etc.

Responses to the reviewers' comments point-by-point

NCOMMS-17-22788A

Thanks a lot for the reviewers' comments, which are very useful for us to improve our paper. With regard to reviewers' comments and suggestions, we reply as follows:

Reviewer #1 (Remarks to the Author):

In this manuscript, the authors provide convincing molecular and biochemical evidence for a novel pathway of L-2-hydroxyglutarate synthesis in select prokaryotes.

Response: Thanks for your good comments.

--What is the effect of hypoxia on CsiD and LGO expression and activity, and on L2HG generation in P. putida?

Response: Thanks for your good suggestions. Shake flask with different rotational speeds is often used in controlling of the dissolved oxygen state of bacteria. The effect of dissolved oxygen on CsiD and LGO expression and activities in *P. putida* KT2440 were studied in shake flask with different rotational speeds. With the decrease of the rotational speeds, the mRNA levels and activities of CsiD and LGO decreased gradually (**Supplementary Fig. 12a–d**). Similar to the expression and activities of CsiD and LGO, the accumulation of intracellular and extracellular L-2-HG both gradually decreased with the decrease of the rotational speeds (**Supplementary Fig. 11b, c**). Thus, hypoxia induced by low rotational speeds would decrease the expression of CsiD and LGO and the generation of L-2-HG.

--How does intracellular redox state modulate L2HG synthesis by this pathway (i.e., how do changes in GSH/GSSG alter synthesis)?

Response: Thanks a lot for your good question. Ratio of NADH to NAD⁺ is often

used to present the intracellular redox state in bacteria. When *P. putida* KT2440 was cultured in shake flask with different rotational speeds, the ratio of NADH to NAD⁺ was assayed. Intracellular redox state would increase with the decrease of rotational speeds (**Supplementary Fig. 11a**), but generation of L-2-HG decreased with the decrease of rotational speeds (**Supplementary Fig. 11b, c**), indicating that increased intracellular L-2-HG is accompanied by decreased ratio of NADH to NAD⁺.

A short paragraph was added in the “Discussion” section as follows:

“Hypoxia can induce L-2-HG production in mammalian cells and increased intracellular L-2-HG is accompanied by the decreased ratio of NADH to NAD⁺^{9,10}. Ratio of NADH to NAD⁺ in *P. putida* KT2440 would increase with the decrease of rotational speeds (**Supplementary Fig. 11a**), but generation of L-2-HG decreased with the decrease of rotational speeds (**Supplementary Fig. 11b, c**). The expression and activities of CsiD and LGO in *P. putida* KT2440 also decreased with the decrease of the rotational speeds (**Supplementary Fig. 12**). This is different from the situation in mammalian cells but understandable since CsiD is a Fe²⁺/2-KG-dependent dioxygenase requiring oxygen to produce L-2-HG and the presence of oxygen supporting L-2-HG production will decrease the ratio of NADH to NAD⁺.”

--The concluding sentence in the discussion, “L-2-HG should not only be treated as an abnormal metabolite, but should be recognized as an important metabolic intermediate during degradation of some organic compounds,” is an appropriate conclusion based on the data, but hardly a novel concept (cf. your references 11 and 12).

Response: According to your good suggestion, the concluding sentence in the discussion was revised as follows:

“Besides a metabolite that can help mitigate cellular reductive stress and play a physiological role in adaptation to hypoxia, L-2-HG is also an important metabolic intermediate in the catabolism of several organic compounds.”

Reviewer #2 (Remarks to the Author):

The authors have discovered a new pathway for the breakdown of L-lysine. The key enzyme of this pathway is CsiD. They show this by proving the GDH is not indispensable for glutarate utilization, that CsiD is induced during glutarate utilization, Knocking out both GDH and CsiD rendered the strain unable to grow on glutarate, They show CsiD is a Fe²⁺/2-KG-dependent deoxygenase producing L-2-HG.

The document shows a novel pathway, containing a new enzyme. The document is well written and deserves publication in Nature communications. A few things have to be changed though.

Response: Thanks for your good comments. We have made corresponding changes according to your suggestions.

Major comments

line 111: why shift from E. coli to P. putida? Please explain.

Response: Thanks for your good question. The aim of this research was to find potential proteins involved in glutarate utilization of *P. putida* KT2440. We have added a short sentence to explain why we shift from *E. coli* to *P. putida* as follows:

“There is a possible unidentified glutarate metabolic pathway in *P. putida* KT2440, and the ortholog proteins of CsiD and YgaF have also been annotated in its genome, named CsiD (PP2909) and LGO (PP2910). Thus, we used *P. putida* KT2440 as a model strain to study the hypothetical role of CsiD and LGO in glutarate utilization.”

The phylogenetic analysis should be part of the results section.

Response: We have moved the phylogenetic analysis into the results section

according to your good suggestion.

The phylogenetic analysis revealed many homologues, but that doesn't prove that they have the same function. Showing enzymatic activity is necessary to prove this. The scope of the paper should therefore be changed to *P. putida* only, general proof to allow "bacteria" as scope is not delivered.

Response: We overexpressed, purified, and biochemically characterized three CsiD and LGO homologues from *E. coli* K-12 MG1655, *Klebsiella pneumoniae* ATCC25955 and *Salmonella enterica* serovar Typhimurium CT18. The results showed that all three homologues of CsiD can catalyze the hydroxylation of glutarate, and the homologues of LGO can catalyze the L-2-HG to produce 2-KG (**Supplementary Table 3**). It is difficult for us to confirm the function of all homologues in the phylogenetic analyses and we have revised the entire manuscript to make the scope of the paper more accurate accordingly.

Please mention the EC number of the enzyme

Response: The EC number of the non-haem Fe²⁺/2-KG-dependent dioxygenase family is EC 1.14.11 according to the reference 40 in the revised manuscript. We have added the EC number of the enzyme as follows:

“CsiD belongs to the non-haem Fe²⁺/2-KG-dependent dioxygenase family (EC 1.14.11)³⁸⁻⁴⁰.”

In the section “References”:

40. Widderich, N. *et al.* Molecular dynamics simulations and structure-guided mutagenesis provide insight into the architecture of the catalytic core of the ectoine hydroxylase. *J. Mol. Biol.* **426**, 586–600 (2014)

Line 45-50, 62-65. The link to pathogenesis is too strong in both introduction and discussion section. It suggests the authors are looking for a cure.

Response: Thanks for your good suggestion. We have deleted some sentences about pathogenesis in both introduction and discussion section according to your suggestion.

Minor comments

Title: The authors have shown that the new pathway is active in *Pseudomonas putida* kt2442, not in other bacteria. Limit the scope of the title.

Response: According to your kind advice, we selected "L-2-Hydroxyglutarate anabolism and catabolism: a metabolic alternative to glutaryl-CoA dehydrogenation pathway in *Pseudomonas putida* KT2440" as the title.

Line 33. Remove "then"

Response: The word has been deleted in the revised manuscript.

line 34: change "it can produce" into "it produced"

Response: We have changed "it can produce" into "it produced" in the revised manuscript.

Line 35: "bacterial" has a too wide scope.

Response: According to your kind advice, we revised the abstract to delineate that all of the experimental data is limited to the specific *Pseudomonas putida* KT2440 as follows:

“Overall, we uncovered that L-2-hydroxyglutarate anabolism and catabolism is a metabolic alternative to the glutaryl-CoA dehydrogenation pathway in *P. putida* KT2440”.

Line 73-76: not necessary, please remove

Response: We have removed the sentences in line 73-76.

Line 84: "microbial" has a too wide a scope.

Response: We revised the introduction to delineate that all of the experimental data is limited to the specific *Pseudomonas putida* KT2440 as follows:

“Based on the results of this study, we uncover an updated version of glutarate and L-2-HG metabolism in *P. putida* KT2440, and reveal that L-lysine can also be viewed as a glucogenic amino acid.”

Line 107: reference 45 does not mention CsiD. Please explain how you come to the conclusion that CsiD is a non-haem FE2+ etc....

Response: Thanks for your reminding. We are sorry that we made such a mistake. We have added a new reference in the revised manuscript to explain this point. The following sentence in the reference could prove the relationship: “Gab protein (CsiD) is encoded by the *csiD* gene (sometimes annotated as *ygaT*) and was identified as a nonhaem FeII-dependent oxygenase (Chance et al., 2002).” The detailed revision of manuscript is listed as follows:

“CsiD belongs to the non-haem Fe²⁺/2-KG-dependent dioxygenase family (EC 1.14.11)³⁸⁻⁴⁰.”

In the section “References”:

38. Lohkamp, B., & Dobritzsch, D. A mixture of fortunes: the curious determination of the structure of Escherichia coli BL21 Gab protein. *Acta Crystallogr. D Biol. Crystallogr.* **64**, 407–415 (2008).

line 295-303. Too speculative

Response: Thanks for your good comment. We agree that the presence of glutarate hydroxylase in humans and plants is rather speculative. We have removed the sentences in line 295-303 according to your suggestion.

Reviewer #3 (Remarks to the Author):

The manuscript present by Zhang and colleges is an interesting metabolic study which uncovers a new metabolic pathway for glutarate metabolism in *P. putida*. Specifically, they identify an Fe²⁺/2-KG-dependent glutarate hydroxylase, unknown so far, encoded by the gene *csiD*. *CsiD* transform glutarate into L-2-Hydroxyglutarate which is further converted in ketoglutarate driven by a new L-2-HG dehydrogenase encoded by the gene *lgo*. Overall, the new pathway yield succinate from glutarate and supposes an alternative pathway to the well-known glutaryl-coenzyme A (CoA) dehydrogenation pathway which yield 2 acetyl-CoA molecules. By knocking out both pathways in *P. putida*, they further construct a glutarate overproducer strain from lysine. They provide enough physiological, genetics, and biochemical experiment supporting these results.

Response: Thanks a lot for your good comments.

Finally, they discuss the presence of both pathways in *P. putida* by means of metabolism optimization. Thus, since the new pathway produce a gluconeogenic final product such as succinate, and the dehydrogenation pathway acetyl-CoA feeding TCA, the simultaneous operation of both pathways should provide a metabolic advantage over strain harboring a single pathway for glutarate degradation. Such results, which constitute one of the more interesting aspects of this manuscript, are poorly supported by direct evidences. Therefore, alternative hypothesis cannot be left aside and the supposed advantage of having two pathways over single pathway should be strongly supported with additional experiments.

Response: We have conducted some experiments to support the competitive advantage of the simultaneous operation of both pathways. The wild-type strain

showed a distinct fitness advantage toward the *P. putida* KT2440 (Δ *csiD*) harboring the glutaryl-CoA dehydrogenation pathway ($W = 20.64 \pm 0.65$) and *P. putida* KT2440 (Δ *gdh*) harboring the glutarate hydroxylation pathway ($W = 5.70 \pm 0.19$).

Major Comment

Along the manuscript glutarate hydroxylation pathway is presented having a competitive advantage toward the dehydrogenation pathway. However the nature of this advantage is not clearly demonstrated. P. putida has high functional redundancy including several ecoparalog enzymes and pathways. Examples include multiple isoenzymes for key TCA steps and multiple catabolic pathways for glucose and amino acids metabolisms. This fact has been explained always in term of metabolic robustness.

Response: Thanks for your good comments. We hope that the evidences provided below could clarify the nature of the competitive advantage and win your satisfaction as possible as we can.

1. The authors clearly demonstrate that *csiD* is strongly induced in the presence of glutarate. What about the expression of genes encoding the dehydrogenase pathway? Are they induced?. According the large lag phase found in the *csiD* knockout, it could be concluded that genes encoding the dehydrogenase pathway are solely induced after a large carbon starving stress and not from the beginning. Ecoparalog or redundant pathways often are deferentially expressed responding to specific environmental conditions. It could be possible that under optimal conditions (those used in the study) only the genes from the hydroxylation pathway are strongly induced?. A time series gene expression analysis of genes from both pathways could clarify these aspects.

Monitoring the gene expression of both pathway under different environmental stresses could be of interest. Also, what about lysine metabolism? Under these conditions (glutarate is produced in the cytoplasm) the behavior of *csiD* and *gdh* knockouts is the same?. Please clarify.

Response: Thanks for your comments. The time series gene expression of both pathways in *P. putida* KT2440 was analyzed by reverse transcription-PCR (RT-PCR). The results showed that both *csiD* and *gdh* were induced at the beginning (0 h) of the growth (**Supplementary Fig. 4a, b**). Additionally, the expressions of *csiD* in *P. putida* KT2440 (Δ *gdh*) and *gdh* in *P. putida* KT2440 (Δ *csiD*) were similar to that of the wild-type strain. Thus, *csiD* in *P. putida* KT2440 (Δ *gdh*) and *gdh* in *P. putida* KT2440 (Δ *csiD*) were also induced at the beginning (0 h) of the growth (**Supplementary Fig. 4c**). The results of quantitative real-time PCR analysis were consistent with that of RT-PCR (**Supplementary Fig. 5**). We also monitored the expression of *csiD* and *gdh* under carbon starvation. The RT-PCR and quantitative real-time PCR analysis showed that both *csiD* and *gdh* were immediately induced under carbon starvation (**Supplementary Fig. 7**).

As for the circumstance of lysine metabolism, the growth curves can be found in **Fig. 5b** of the main text. The wild-type and the two mutants have longer lag phases than cultured in glutarate medium. *P. putida* KT2440 (Δ *csiD*) grows faster than that of *P. putida* KT2440 (Δ *gdh*). Similar to the situation of the cultivation in glutarate, *csiD* and *gdh* were both induced at the beginning of the growth (**Supplementary Fig. 6**).

2. Pg. 11. Fig. 4

"As shown in Fig. 4f, the wild-type strain show a distinct fitness advantage toward

the *P. putida* KT2440 (Δ csiD) harboring the glutaryl-CoA dehydrogenation pathway ($W = 20.64$) and *P. putida* KT2440 (Δ gdh) harboring the glutarate hydroxylation pathway ($W = 5.70$). In addition, *P. putida* KT2440 (Δ gdh) harboring the glutarate hydroxylation pathway was more competitive than *P. putida* KT2440 (Δ csiD) harboring the glutaryl-CoA dehydrogenation pathway ($W = 5.35$), implying that the glutarate hydroxylation pathway provides a competitive advantage over the glutaryl-CoA dehydrogenation pathway during glutarate utilization."

The supposed advantage of hydroxylation pathway addressed by these competition experiments also is questionable. If the dehydrogenation genes are not induced from the beginning, is not possible to conclude whether the advantage is because a metabolic or regulation bottlenecks. Solving the gene expression profile of both pathways as discussed above will help in this topic.

From the Fig. 4a, can be concluded that excluding the lag phase, the growth rate and final biomass yield of csiD knockout is similar than that from the WT strain. Therefore, it's not evident this advantage in term of metabolic optimization but again these results argue in favor of regulatory bottleneck. In other words, the hydrogenation pathway seems as efficient as hydroxylation pathway after complete induction. Exhaustive and quantitative estimation of growth rates, final biomass yields etc for each knockout strain should be done and included in the paper to clarify, even more, the yield of each pathway.

Response: Thanks for your comments. According to your advice, exhaustive and quantitative estimation of growth rates, final biomass yields etc for each knockout strain has been included in the revised manuscript (**Supplementary Table 1**). The growth rate during exponential growth period and the maximum specific growth rate of *P. putida* KT2440 (Δ gdh) are both higher than that of *P. putida* KT2440 (Δ csiD),

demonstrating that the hydroxylation pathway has a competitive advantage toward dehydrogenation pathway.

The expression of both pathways was studied and these results indicated that the dehydrogenation pathway would be induced at the beginning of the growth, implying that the competitive advantage was not caused by the order of each pathway to be induced. To further proving that advantage of hydroxylation pathway is not because a regulation bottleneck, we complemented *csiD* or *gdh* in *P. putida* KT2440 ($\Delta gdh \Delta csiD$). The expression of *csiD* or *gdh* is both controlled by the constitutive promoter P_c or the inducible promoter P_{tac} . The results in **Supplementary Fig. 8** indicated that *csiD* complement strain grows faster than *gdh* complement strain under the control of P_c or P_{tac} . The hydroxylation pathway and dehydrogenation pathway expressed under the same promoter, but the mutant strain harboring the glutarate hydroxylation pathway was more competitive than the mutant strain harboring the glutaryl-CoA dehydrogenation pathway. Therefore, we can conclude that the advantage of hydroxylation pathway is not because a regulation bottleneck but possibly comes from its biochemical character.

3. Fig. 6. The authors discuss a very interesting synergy between both pathways. Thus, while the hydroxylation pathway would provide C4 carbon products, the dehydrogenation pathway would be in charge of providing Acetyl-CoA. Overall, the activation of both pathways would be an advantage over single pathways. In any case, the activation of dehydrogenation pathway has not been demonstrated. The experiments using knockout strains on glyoxylate cycle only provide indirect evidences since they are taken after 12 h of cultivation. This 12 h seems to be a magic number. Maximum OD is achieved by the wt and *gdh* knockout (Fig. 1). The lag

phase in the *csiD* knockouts is around 12 h (Fig.4). 12 h are needed to start the consumption of glutarate in this same knockout strain (Fig.4). What is the behavior of such knockouts strain at early time?. There is not synergy at mid exponential phase?.

Experiment carried out with the *gdh* knockout (Fig. 1) could suggest that the role of dehydrogenation pathway is solely needed at the end of exponential phase. Again, the differential expression of both pathways is on the table.

The synergistic action of several redundant pathways operating in response of different environmental stress is very interesting and widely extended in robust bacteria such as *P. putida*. Much more direct evidence should be desirable here. ^{13}C experiments could elucidate the role of each pathway in glutarate metabolism. In addition, the above indicate gene expression profiles could help.

Response: Thanks for your good suggestion. The expression of both pathways was studied and the activation of dehydrogenation pathway has been demonstrated (**Supplementary Figs. 4 and 5**). Although *P. putida* KT2440 ($\Delta\textit{csiD}$) displayed a slower growth rate and a longer lag time than *P. putida* KT2440 ($\Delta\textit{gdh}$) and wild-type strain, the activation of *gdh* at the beginning of the growth in this mutant has been demonstrated. The expression of *csiD* and *gdh* were synergetic from 0 h to 10 h (including the mid exponential phase) by the approach of quantitative real-time RT-PCR (**Supplementary Fig. 5**). These results indicated that the dehydrogenation pathway would be induced at the beginning of the growth, which can conclude that the advantage of hydroxylation pathway is not due to a regulation bottleneck.

According to your kind suggestions, tracing experiment using ^{13}C -labeled glutarate can directly elucidate the role of each pathway in glutarate metabolism. We contacted all of the isotopic reagents companies (such as Cambridge Isotope Laboratories, Medical Isotope, Inc., Omicron Biochemicals, Inc.) to purchase the U- ^{13}C -labeled

glutarate. However, there is no feasible approach to buy U-¹³C-labeled glutarate at this stage. Thus, we are very sorry for that it is not available for us to perform this experiment.

A short paragraph was added in the “Discussion” section as follows:

“Time series gene expression analysis indicated that both *csiD* and *gdh* were induced at the beginning of growth using glutarate as the sole carbon source (**Supplementary Figs. 4 and 5**). When expressed under the same promoter, *csiD* could support faster growth of *P. putida* KT2440 (Δ *gdh* Δ *csiD*) than that of *gdh* (**Supplementary Fig. 8**). Succinate, the end product of glutarate hydroxylation pathway, is an intermediate of TCA cycle and can be used more quickly than crotonyl-CoA, the intermediate of glutaryl-CoA dehydrogenation pathway. The competitive advantage of glutarate hydroxylation pathway is not caused by its immediate expression but possibly due to its ability to provide the quickly utilizable metabolic intermediate, succinate. This notion might be further addressed by tracing experiment using ¹³C-labeled glutarate.”

4. This reviewer is fully convinced of the need to increase biochemical studies providing new enzymatic activities and metabolic pathways. The advent of genomics has overshadowed detailed biochemistry studies and, as a fatal consequence, new activities are now assigned, in many cases, based on bioinformatics evidences with low confidence. For this reason, I considered that the new activities found here have a great value. However, since the activity of CsiD has been reported here for the first time, I miss basic biochemical characterization of this interesting enzyme. The authors should consider providing this information. At least km for each substrate, Vmax, kcat etc.

Response: Thanks a lot for your good suggestion. The basic biochemical characterization of CsiD from *P. putida* has been assayed and included in the revised manuscript (**Table 1**).

Reviewers' comments:

Reviewer #1 (Remarks to the Author):

None.

Reviewer #2 (Remarks to the Author):

The authors have satisfactorily responded to my earlier concerns. I propose to accept the paper for publication.

Reviewer #3 (Remarks to the Author):

The revised work of Zhang et al is meritorious, and potentially publishable. However, several technical mistakes which could lead to wrong conclusions should be addressed in order to increase the value of the current manuscript.

1. “The CsiD-dependent oxygen consumption detected by a Clark-type oxygen electrode was almost absent without the addition of 2-KG, while noticeable oxygen consumption was observed when 2-KG was added (Fig. 2d), indicating that CsiD is a Fe²⁺/2-KG-dependent dioxygenase.”

The oxygen consumption in the assay with 2-KG in absence of glutarate should be considered as unspecific activity. In any case, this is not a probe that LhgO is an Fe²⁺/2-KG-dependent dioxygenase. To show this, the assay should be done including glutarate in absence of 2-KG as the authors show in fig. 2.f. Please rephrase this sentence accordingly.

2. From figure 2.d, could be estimate the O₂/glutarate stoichiometry?

3. “It was also found that the activity of CsiD was reduced to 80.3% and 42.6% in the absence of Fe²⁺ or ascorbate, respectively (Fig. 2f). Thus, CsiD also requires these cofactors for enzymatic activity as other standard Fe²⁺/2-KG-dependent dioxygenases”.

Unfortunate statement. In any case ascorbate is a cofactor in the catalysis. Ascorbate is a reducing agent used to keep Fe²⁺ reduced and avoiding its reduction to Fe³⁺ in the presence of oxygen. Thus, this is the reason of the activating effect upon ascorbic acid addition in the assay. Please, rephrase this sentence.

4. “2-HG exists in two stereoisomeric forms: L-2-HG and D-2-HG^{2,4}. As shown in Fig. 2h, D-2-HG was efficiently catalyzed by D-2-HG dehydrogenase (HGDH) from the anaerobic bacterium *Acidaminococcus fermentans*, whereas L-2-HG was not under the same assay conditions. When reaction products of CsiD were added to the assay mixture, no HGDH activity was detected (Fig. 2h), indicating that the product obtained from glutarate by CsiD is L-2-HG, instead of D-2-HG”.

How HGDH activity was performed? This information is missing in methods section.

5. PP_2910 is annotated in the genome of *P. putida* as *lhgO* (not LGO). Since PP_2910 is 71% identical to YgaF (*lhgO*) of *E. coli*, I suggest using along the manuscript *lhgO* (*LhgO*) gene (protein) instead of LGO.

6. “The ability of LGO to use oxygen as a direct electron acceptor was assessed using a Clark-type oxygen electrode. As shown in Fig. 3d, in the absence of any organic electron acceptor, the consumption of oxygen was barely detectable. In contrast, upon addition of 1 mM phenazine methosulfate (PMS), there was a rapid depletion of oxygen in the enzyme reaction mixture. These data unequivocally establish LGO as a dehydrogenase with very poor reactivity with molecular oxygen”.

YgaF was identified as a flavoprotein with L-2-hydroxyglutarate oxidase activity (ref 44). Why the authors indicate that LGO is an L-2-HG dehydrogenase? FAD group in YgaF is not covalently bound to the protein so, we cannot rule out the possibility that the FAD cofactor in *P. putida* LGO is lost during the purification steps explaining, in some extent the absent of oxygen consumption in absence of electron acceptors. Please perform the LGO oxidase activity in the presence of FAD(H) or provide further evidences supporting the LGO dehydrogenase activity.

7. “The expression of *csiD* and *gdh* during growth in the glutarate medium was analyzed by RT-PCR and quantitative real-time PCR (qPCR). Both *csiD* and *gdh* were induced at the beginning of growth in *P. putida* KT2440. *csiD* in *P. putida* KT2440 (Δ *gdh*) and *gdh* in *putida* KT2440 (Δ *csiD*) were also immediately expressed at the beginning of the growth (Supplementary Figs. 4 and 5).”

How the experiments have been performed? What it means 0h?, how the precultures were done?. It's difficult to think that at 0h, before to be in contact with glutarate, both *csiD* and *gdh* exhibit as high expression level as in exponential growth phase. If the preculture were done in rich medium, the authors should have into account that there is a very high level of lysine in standard LB medium, so it could explain these high expression levels of *csiD* and *gdh* at early times. By the way, qPCR is a useful approach estimating relative expression levels, but the inclusion of a well-known housekeeping gene is mandatory. At the best of my knowledge, there is not any

well-accepted housekeeping gene in *P. putida*. In any case, the use of 16 sRNA gene is completely inappropriate in this case as this is a gene which expression is growth-phase dependent. Thereof, using this gene as reference could lead wrong results. I recommend performing a quantitative qPCR by using a calibration curve with the same PCR amplicons.

8. “The expression and activities of CsiD and LGO in *P. putida* KT2440 also decreased with the decrease of the rotational speeds (Supplementary Fig. 12).”

Again, the use of 16 sRNA gene as reference largely overcome the conclusions from figure 12.a y 12.b. See comment 7

Also, units in figures 12.c and 12.d are U/ml, what these units means? ml of culture? Higher rotational speeds yield higher biomass/ml, so this is not the right way to quantify these enzymatic activities. Please clarify.

9. “Additionally, *csiD* and *gdh* were both induced at the beginning of the utilization of L-lysine (Supplementary Fig. 6) and carbon starvation (Supplementary Fig. 7)”.

The expression of LGO in *E. coli* was found induced by RpoS during carbon starvation and at stationary phase (ref 44). Despite RT-PCR is not the best approach to finely quantify gene expression (a completely quantitative qPCR experiments would be diserable), according figure 7 it seems that *csiD* (and probably LGO), are induced during carbon starving conditions, however, under stationary phase they are strongly repressed (fig 4, 5). Could the authors discuss this different behavior between *E. coli* and *P. putida*.

Responses to the reviewers' comments point-by-point

NCOMMS-17-22788B

Thanks a lot for the reviewers' comments. With regard to reviewers' comments and suggestions, we reply as follows:

Reviewer #1 (Remarks to the Author):

None.

Response: Thank you very much for your time on our paper.

Reviewer #2 (Remarks to the Author):

The authors have satisfactorily responded to my earlier concerns. I propose to accept the paper for publication.

Response: Thank you very much for your time on our paper.

Reviewer #3 (Remarks to the Author):

The revised work of Zhang et al is meritorious, and potentially publishable. However, several technical mistakes which could lead to wrong conclusions should be addressed in order to increase the value of the current manuscript.

Response: Thanks a lot for your valuable comments. The experiments related to the expression of *csiD* and *gdh* have been conducted with preculture in MSM with glucose as the sole carbon source and quantitative qPCR by using a calibration curve. The same results as obtained in previous experiments were acquired. We also have made other corresponding changes according to your suggestions. We hope that these changes will enable our manuscript to win your satisfaction.

1. "The CsiD-dependent oxygen consumption detected by a Clark-type oxygen

electrode was almost absent without the addition of 2-KG, while noticeable oxygen consumption was observed when 2-KG was added (Fig. 2d), indicating that CsiD is a Fe²⁺/2-KG-dependent dioxygenase.”

The oxygen consumption in the assay with 2-KG in absence of glutarate should be considered as unspecific activity. In any case, this is not a probe that LhgO is an Fe²⁺/2-kg-dependent dioxygenase. For show this, the assay should be done including glutarate in absence of 2-KG as the authors show in fig. 2.f. Please rephrase this sentence accordingly.

Response: Thanks for your good suggestion. According to your suggestion, the assay has been done including glutarate in absence of 2-KG (added in revised **Fig. 2c**). No oxygen consumption was observed in the assay with glutarate in absence of 2-KG. As you mentioned above, the oxygen consumption in the assay with 2-KG in absence of glutarate should be considered as unspecific activity. We have rephrased the sentence accordingly as follows:

“The CsiD-dependent oxygen consumption detected by a Clark-type oxygen electrode was almost absent without the addition of 2-KG, while noticeable oxygen consumption was observed when 2-KG was added (**Fig. 2c**). The product of the CsiD-catalyzed reaction toward 2-KG was analyzed by high-performance liquid chromatography (HPLC). A compound that had a retention time of 19.37 min, which corresponded to the peak of authentic succinate, was detected (**Fig. 2d**). The result of liquid chromatography-tandem mass spectrometry (LC-MS/MS) further confirmed the production of succinate by CsiD (**Supplementary Fig. 1a**). Like other typical Fe²⁺/2-KG-dependent dioxygenases, CsiD has unspecific activity toward 2-KG to produce succinate.”

2. From figure 2.d, could be estimate the O₂/glutarate stoichiometry?

Response: Thanks for your comments. The experiment in Fig. 2d (revised **Fig. 2c**) was conducted with a Clark-type oxygen electrode (Oxytherm, Hansatech, United Kingdom), through which only oxygen consumption could be assayed. Thus, the O₂/glutarate stoichiometry could not be estimated from Fig. 2d.

In almost all cases, Fe²⁺/2-KG-dependent dioxygenases require two oxygen atoms for completing the hydroxylation of their substrates. During catalysis, the substrate accepts one oxygen atom while 2-KG undergoes a decarboxylation reaction consuming the remaining oxygen atom to form succinate and CO₂ (Martinez S and Hausinger RP, Catalytic mechanisms of Fe(II)-and 2-oxoglutarate-dependent oxygenases, J Biol Chem. 2015 290:20702-11). Thus, the O₂/glutarate stoichiometry of CsiD catalyzed glutarate hydroxylation is mostly likely to be 1:1.

3. “It was also found that the activity of CsiD was reduced to 80.3% and 42.6% in the absence of Fe²⁺ or ascorbate, respectively (Fig. 2f). Thus, CsiD also requires these cofactors for enzymatic activity as other standard Fe²⁺/2-KG-dependent dioxygenases”.

Unfortunate statement. In any case ascorbate is a cofactor in the catalysis. Ascorbate is a reducing agent used to keep Fe²⁺ reduced and avoiding its reduction to Fe³⁺ in the presence of oxygen. Thus, this is the reason of the activating effect upon ascorbic acid addition in the assay. Please, rephrase this sentence.

Response: Thanks for your good suggestion. We have rephrased the sentence as follows:

“Ascorbate is an established activator of Fe²⁺/2-KG-dependent dioxygenases through maintaining Fe²⁺ in the reduced state^{42,43}. The activity of CsiD was reduced to

42.6% in the absence of ascorbate (**Fig. 2f**). Additionally, the activity of CsiD was reduced to 80.3% in the absence of Fe^{2+} , suggesting that CsiD requires Fe^{2+} for enzymatic activity as other standard Fe^{2+} /2-KG-dependent dioxygenases⁴⁴.”

4. “2-HG exists in two stereoisomeric forms: L-2-HG and D-2-HG^{2,4}. As shown in Fig. 2h, D-2-HG was efficiently catalyzed by D-2-HG dehydrogenase (HGDH) from the anaerobic bacterium *Acidaminococcus fermentans*, whereas L-2-HG was not under the same assay conditions. When reaction products of CsiD were added to the assay mixture, no HGDH activity was detected (Fig. 2h), indicating that the product obtained from glutarate by CsiD is L-2-HG, instead of D-2-HG”.

How HGDH activity was performed? This information is missing in methods section.

Response: Thanks for your reminding. The method of HGDH activity measurement has been included in the revised Supplementary Methods section as follows:

“The gene encoding NAD^+ -dependent D-2-hydroxyglutarate dehydrogenase (HGDH) from *A. fermentans* was synthesized by General Biosystems, Inc. (Anhui, China) and ligated into expression plasmid pETDuet-1. The expression and purification procedures of HGDH were the same as CsiD and LhgO of *P. putida* KT2440. The activity of HGDH was measured in 100 μL solution containing 75 μL assay solution and 25 μL sample. The assay solution contained 100 mM HEPES (pH 8.0), 100 μM NAD^+ , 0.1 μg HGDH, 5 μM resazurin and 0.01 U mL^{-1} diaphorase (Sigma-Aldrich, USA). After 75 μL assay solution was added to 25 μL sample and incubated in the dark for 30 min in black 96-well plates (PerkinElmer, USA), fluorescence was measured using a fluorescence microplate reader (PerkinElmer,

USA) with excitation at 540 ± 10 nm and emission of 610 ± 10 nm.”

5. PP_2910 is annotated in the genome of *P. putida* as *lhgO* (not LGO). Since PP_2910 is 71% identical to *YgaF* (*lhgO*) of *E. coli*, I suggest using along the manuscript *lhgO* (*LhgO*) gene (protein) instead of LGO.

Response: Thanks a lot for your good suggestion. We have changed *lgo* (LGO) gene (protein) to *lhgO* (*LhgO*) throughout the manuscript.

6. “The ability of LGO to use oxygen as a direct electron acceptor was assessed using a Clark-type oxygen electrode. As shown in Fig. 3d, in the absence of any organic electron acceptor, the consumption of oxygen was barely detectable. In contrast, upon addition of 1 mM phenazine methosulfate (PMS), there was a rapid depletion of oxygen in the enzyme reaction mixture. These data unequivocally establish LGO as a dehydrogenase with very poor reactivity with molecular oxygen”.

YgaF was identified as a flavoprotein with L-2-hydroxyglutarate oxidase activity (ref 44). Why the authors indicate that LGO is an L-2-HG dehydrogenase? FAD group in *YgaF* is not covalently bound to the protein so, we cannot rule out the possibility that the FAD cofactor in *P.putida* LGO is lost during the purification steps explaining, in some extent the absent of oxygen consumption in absence of electron acceptors. Please perform the LGO oxidase activity in the presence of FAD(H) or provide further evidences supporting the LGO dehydrogenase activity.

Response: Thanks for your good suggestion. We purified *YgaF* from *E. coli* K-12 and *LhgO* from *P. putida* KT2440, incubated them with FAD or FMN, and studied their activities with a Clark-type oxygen electrode. Just as you supposed, the oxidase activity of *LhgO* can be stimulated by FAD, indicating that part of FAD cofactor was

lost during the purification of LhgO. In addition, the oxygen consumption rate of LhgO was faster than that of YgaF at the same protein concentration (**Fig. 3f**). Thus, like YgaF from *E. coli* K-12, LhgO from *P. putida* KT2440 is a FAD-dependent L-2-hydroxyglutarate oxidase. Thanks a lot for your constructive suggestion, and we have added the data mentioned above in the revised manuscript and renamed LhgO as an L-2-hydroxyglutarate oxidase.

7. “The expression of *csiD* and *gdh* during growth in the glutarate medium was analyzed by RT-PCR and quantitative real-time PCR (qPCR). Both *csiD* and *gdh* were induced at the beginning of growth in *P. putida* KT2440. *csiD* in *P. putida* KT2440 (Δ *gdh*) and *gdh* in *putida* KT2440 (Δ *csiD*) were also immediately expressed at the beginning of the growth (Supplementary Figs. 4 and 5).”

How the experiments have been performed? What it means 0h?, how the precultures were done?. It’s difficult to think that at 0h, before to be in contact with glutarate, both *csiD* and *gdh* exhibit as high expression level as in exponential growth phase. If the preculture were done in rich medium, the authors should have into account that there is a very high level of lysine in standard LB medium, so it could explain these high expression levels of *csiD* and *gdh* at early times. By the way, qPCR is a useful approach estimating relative expression levels, but the inclusion of a well-known housekeeping gene is mandatory. At the best of my knowledge, there is not any well-accepted housekeeping gene in *P. putida*. In any case, the use of 16 sRNA gene is completely inappropriate in this case as this is a gene which expression is growth-phase dependent. Thereof, using this gene as reference could lead wrong results. I recommend performing a quantitative qPCR by using a calibration curve with the same PCR amplicons.

Response: Thanks a lot for your suggestions. In the previous experiments, *P. putida* KT2440 was grown overnight in LB medium, collected by centrifugation, washed with normal saline and then inoculated into MSM with 5 g L⁻¹ glutarate as the sole carbon source. After inoculated into the MSM with 5 g L⁻¹ glutarate as the sole carbon source, cells of *P. putida* KT2440 were immediately harvested and termed as the sample at 0 h. The expression of *csiD* and *gdh* at 0 h in the previous experiments might be due to the fact that *P. putida* KT2440 has already been in contact with glutarate at that time.

Just as you mentioned, the expression of *csiD* and *gdh* may be influenced by lysine in the rich medium during the precluture process. Thus, *P. putida* KT2440 was grown overnight in LB medium, collected by centrifugation, washed with normal saline and then inoculated into MSM with 5 g L⁻¹ glucose as the sole carbon source. Cells of *P. putida* KT2440 in MSM with 5 g L⁻¹ glucose as the sole carbon source were harvested at the exponential growth phase, washed three times and inoculated into MSM with 5 g L⁻¹ glutarate as the sole carbon source. Cells of *P. putida* KT2440 were immediately harvested and termed as the sample at 0 h.

We also used absolute quantitative real-time PCR to quantify the copy number of *csiD* and *gdh* according to your suggestion. The recombinant plasmids Blunt-*csiD* and Blunt-*gdh* with the same PCR amplicons of *csiD* and *gdh* were constructed. These two plasmids were extracted using a Plasmid Miniprep Kit (Biomiga, USA) and quantified by NanoDrop ND-1000 (Thermo Scientific, USA). The copy numbers of the plasmids were calculated based on the molecular weight of the recombinant plasmids. A 10-fold serial dilution series of the recombinant plasmids, ranging from 1 × 10³ to 1 × 10⁸ copies μL⁻¹, was used to construct the standard curves for *csiD* and *gdh*. Data analysis was carried out with OriginPro software 8.0 (OriginLab, USA)

(Supplementary Fig. 13).

After preparation of the samples using the new procedure and quantitative qPCR by using a calibration curve, the same results as obtained in previous experiments were acquired. Both *csiD* and *gdh* were induced immediately when the strain was in contact with glutarate (**Supplementary Fig. 4 and Fig. 5**).

8. “The expression and activities of CsiD and LGO in *P. putida* KT2440 also decreased with the decrease of the rotational speeds (Supplementary Fig. 12).”

Again, the use of 16 sRNA gene as reference largely overcome the conclusions from figure 12.a y 12.b. See comment 7

Also, units in figures 12.c and 12.d are U/ml, what these units means? ml of culture? Higher rotational speeds yield higher biomass/ml, so this is not the right way to quantify these enzymatic activities. Please clarify.

Response: Thanks for your good suggestion. After preparation of the samples using the new procedure and quantitative qPCR by using a calibration curve, the expression of *csiD* and *lhgO* at different rotational speeds were reanalyzed. The same results as obtained in previous experiments were acquired (**Supplementary Fig. 12a, b**).

As for the activities of CsiD and LhgO, we are very sorry for our unprofessional use of the unit in previous manuscript. According to your advice, we have changed the unit to U mg⁻¹ and the related experiments were conducted and clarified in revised manuscript as follows:

“*P. putida* KT2440 was grown to mid-log stage (OD₆₀₀ = 2) in 50 mL glutarate medium at different rotational speeds. Cells were harvested by centrifugation (6,000 × g for 10 min at 4 °C), washed and suspended in phosphate-buffered saline (PBS) supplemented with 1 mM phenylmethylsulfonyl fluoride (PMSF). The final OD₆₀₀ of

the cells was 20 and the cells were disrupted by sonication with a Sonics sonicator (500 W; 20 KHz) in an ice bath. The cell debris were removed by centrifugation at $13,000 \times g$ for 5 min at 4 °C and the resultant supernatants were used as the crude cell extracts to measure the activities of CsiD and LhgO. The protein concentration of the crude cell extracts was determined by the method of Bradford, with bovine serum albumin as a standard.”

9. “Additionally, *csiD* and *gdh* were both induced at the beginning of the utilization of L-lysine (Supplementary Fig. 6) and carbon starvation (Supplementary Fig. 7)”.

The expression of LGO in *E. coli* was found induced by RpoS during carbon starvation and at stationary phase (ref 44). Despite RT-PCR is not the best approach to finely quantify gene expression (a completely quantitative qPCR experiments would be desirable), according figure 7 it seems that *csiD* (and probably LGO), are induced during carbon starving conditions, however, under stationary phase they are strongly repressed (fig 4, 5). Could the authors discuss this different behavior between *E. coli* and *P. putida*..

Response: Thanks a lot for your suggestions. The expression of *csiD* in *E. coli* was found to be activated by RpoS during carbon starvation and at stationary phase (ref 44, Mol Microbiol. 2004, 51:799-811). CsiR, a repressor that acts at the *csiD* promoter, was also identified in this reference. CsiR is likely to be allosterically regulated by some small compound that is produced during carbon starvation or cells enter into stationary phase (ref 44, Mol Microbiol. 2004, 51:799-811). The expression of *csiD* requires both RpoS and an effector to release CsiR from the promoter of *csiD*. However, the effector of CsiR has never been identified. The ortholog protein of CsiR (PP2908) has been annotated upstream of CsiD (PP2909) in genome of *P. putida*

KT2440. Since expression of *csiD* in *P. putida* KT2440 was induced by glutarate, it can be speculated that *csiD* in *P. putida* KT2440 is also regulated by CsiR and glutarate is the effector of CsiR. In both *E. coli* and *P. putida* KT2440, the expression of CsiD is induced during carbon starvation, which might due to the allosteric regulation of CsiR by glutarate produced from lysine degradation. Different from the situation in ref 44, *P. putida* KT2440 was cultured in MSM containing glutarate as the sole carbon source in our experiments. When *E. coli* entered into stationary phase in rich medium and under carbon starvation conditions, glutarate may begin to accumulate and induce the expression of *csiD*. However, when *P. putida* KT2440 entered into stationary phase in glutarate mudium, glutarate added in medium has been depleted and no effector of CsiR is present, leading to the repression of the *csiD* expression.

A short discussion has been added in the revised manuscript as follows:

“The expression of *csiD* is activated by RpoS and repressed by CsiR in *E. coli*^{36,37,56}. CsiR is allosterically regulated by some small compound produced during carbon starvation or when cells enter into stationary phase. CsiD and YgaF were supposed to be involved in production or catabolism of this compound³⁷. The ortholog protein of CsiR (PP2908) locates upstream of CsiD (PP2909) in genome of *P. putida* KT2440. The expression of *csiD* and *lhgO* in *P. putida* KT2440 was induced by carbon starvation or exogenously added glutarate but repressed under stationary phase when glutarate added in medium had been depleted. CsiD and LhgO were found to construct a hydroxylation pathway for catabolism of glutarate. Thus, the expression of *csiD* and *lhgO* may also be regulated by CsiR in *P. putida* KT2440. Glutarate, a compound that can be produced from lysine degradation during carbon starvation or exogenously added, might be the effector of CsiR. The molecular function of CsiR

and its regulation mechanism deserve further investigation.”

Reviewers' Comments:

Reviewer #3 (Remarks to the Author):

The authors have done a great effort responding my previous concerns. The resulting work is meritorious and I dont have additional comments. I propose to accept the paper for publication. Good job.

Responses to the referees' comments point-by-point

NCOMMS-17-22788C

Thanks a lot for the referees' comments. With regard to referees' comments and suggestions, we reply as follows:

Reviewer #3 (Remarks to the Author):

The authors have done a great effort responding my previous concerns. The resulting work is meritorious and I dont have additional comments. I propose to accept the paper for publication. Good job.

Response: Thank you very much for your time on our paper.